# Pangenome graphs improve the analysis of structural variants in rare genetic diseases

Cristian Groza [1], Carl Schwendinger-Schreck[2], Warren A. Cheung [2], Emily G. Farrow [2], Isabelle Thiffault [2], Juniper Lake [3], William B. Rizzo[4], Gilad Evrony [5], Tom Curran [6], Guillaume Bourque [7,8,9,10] ✉ & Tomi Pastinen [2] ✉

Rare DNA alterations that cause heritable diseases are only partially resolvable by clinical next-generation sequencing due to the difficulty of detecting structural variation (SV) in all genomic contexts. Long-read, high fidelity genome sequencing (HiFi-GS) detects SVs with increased sensitivity and enables assembling personal and graph genomes. We leverage standard reference genomes, public assemblies ($n = 94$) and a large collection of HiFi-GS data from a rare disease program (Genomic Answers for Kids, GA4K, $n = 574$ assemblies) to build a graph genome representing a unified SV callset in GA4K, identify common variation and prioritize SVs that are more likely to cause genetic disease (MAF < 0.01). Using graphs, we obtain a higher level of reproducibility than the standard reference approach. We observe over 200,000 SV alleles unique to GA4K, including nearly 1000 rare variants that impact coding sequence. With improved specificity for rare SVs, we isolate 30 candidate SVs in phenotypically prioritized genes, including known disease SVs. We isolate a novel diagnostic SV in *KMT2E*, demonstrating use of personal assemblies coupled with pangenome graphs for rare disease genomics. The community may interrogate our pangenome with additional assemblies to discover new SVs within the allele frequency spectrum relevant to genetic diseases.

Structural variants (SVs) contribute to Mendelian and complex disease, yet they are the most challenging to detect, assemble, and fully resolve. Indeed, many SVs are in repetitive sequences that are difficult to approach with short-read sequencing libraries[1,2]. In contrast, long-reads can detect and characterize much more complex and repetitive structural variants[3] and can be used to efficiently build reference-free de novo assemblies from genomes as large as a human genome[4]. Long-reads, together with information from parental sequencing, also enable

the phased assembly of haplotype-resolved maternal and paternal genomes based on unique k-mers[5]. Previous efforts employing long-reads have discovered up to 28,000 SVs per human genome[6]. However, we still need to adopt computational methods to fully leverage this richer data in the context of rare diseases. While SV callers that operate on whole genome assemblies exist[7,8], their approach of comparing a proband genome against a single reference genome may fail, even with high-quality genome assemblies, since some regions may be

[1]Quantitative Life Sciences, McGill University, Montréal, QC, Canada. [2]Genomic Medicine Center, Children's Mercy Hospital and Research Institute, KC, MO, USA. [3]Pacific Biosciences, Menlo Park, CA, USA. [4]Child Health Research Institute, Department of Pediatrics, Nebraska Medical Center, Omaha, NE, USA. [5]Center for Human Genetics and Genomics, Department of Pediatrics, Neuroscience & Physiology, New York University Grossman School of Medicine, New York, NY, USA. [6]Children's Mercy Research Institute, Kansas City, MO, USA. [7]Canadian Center for Computational Genomics, McGill University, Montréal, QC, Canada. [8]Department of Human Genetics, McGill University, Montréal, QC, Canada. [9]Institute for the Advanced Study of Human Biology (WPI-ASHBi), Kyoto University, Kyoto, Japan. [10]Victor Phillip Dahdaleh Institute of Genomic Medicine at McGill University, Montréal, QC, Canada. ✉e-mail: guil.bourque@mcgill.ca; tpastinen@cmh.edu

absent or contain very different alleles. Moreover, it remains difficult to compare alleles among genomes since the genomes are only related to the reference genome and not to each other. While tools exist to call and cluster SVs[9–11], they rely on heuristics such as the maximum distance between events or proximity graphs[12], which may erroneously split or merge SVs because they do not consider the entire genome assembly or variation between alleles in complex loci. Therefore, accurate estimation of SV allele frequency may benefit from tools that align and compare the various alleles observed in complex loci.

To alleviate these shortcomings and fully leverage high-quality genome assemblies, a pangenomic approach, where genomes are related to each other in a graph, is necessary. Some approaches have been developed to achieve this through progressive alignment of genome assemblies[13,14], while others do so through pairwise comparisons[15]. The resulting pangenome graphs, built from high-quality genome assemblies, have already been used to create a pangenome reference of human population diversity by the Human Pangenome Reference Consortium (HPRC)[16]. This new type of reference showed increased sensitivity in detecting SVs over methods that use a linear reference. Pangenome graphs also provide a unified SV callset where the boundaries of polymorphisms are delimited by bubbles, and the alleles are precisely defined as paths through bubbles. This allows for more robust allele frequencies, especially in the case of multiallelic SVs.

Here, we explore the benefits of using such a strategy to characterize structural variation in a rare disease cohort, exclude common non-pathogenic or infrequent (MAF 1–5%) variation, and prioritize SVs that are sufficiently rare to be causal (MAF < 0.01). Also, we show how pangenome methods can be used along with other tools to improve sensitivity and specificity in detecting SVs.

## Results

### A pangenomic approach to identify and integrate structural variation across hundreds of genomes

We pursued the discovery of rare SVs that were potentially pathogenic among a cohort of 287 parent–offspring trios included in the Genomic Answers for Kids (GA4K) program targeting pediatric genetic disease[17]. In this cohort, prior to assembling the genomes, more than 90% of the probands remained undiagnosed after chromosomal microarray analysis or even standard clinical sequencing and systematic exploration of putatively causative single nucleotide variants (Supplementary Data 1), with only less than 10% being eventually diagnosed. Thus, this set of genomes is enriched for difficult to solve cases. We included short-read genome sequencing (srGS) parental data and further sequenced all the probands using PacBio HiFi reads (Methods) at a mean depth of 27× (Fig. 1A, median 27×, range 6–48×). A subset of this HiFi-GS data, but none of the assemblies, was included in an earlier study[17]. Here, we expanded the cohort and systematically developed assemblies of 574 haploid proband genomes using hifiasm[5], obtaining a mean N50 of 18.2 Mbp (Fig. 1B, median 16.4 Mbp, range 78.6 kbp–55.3 Mbp). To facilitate the identification of rare variants, we also augmented our data with the 94 haploid genomes released by the HPRC[16]. We then created a pangenome graph with minigraph[13], which was previously tested and found to highly agree with reference-based methods[16], to identify structural variants in the combined set of 668 haploid genomes together with two standard reference genomes (GRCh38 and CHM13v2). We chose minigraph since it scales linearly with the number of genomes at the expense of requiring a backbone reference to build the graph. During graph construction, when a haploid genome was added, polymorphisms that were larger than 50 base pairs (bp) but missing from the graph created new nodes and paths (Fig. 1C). With our data, we found that the number of new non-reference sequence nodes added from each additional haploid genome plateaued at around 500 (Fig. 1D). This suggests that there remains many more alleles to be discovered in human genomes, continuing the trend previously observed in the HPRC dataset[16].

Using the resulting graph, we genotyped the assemblies and observed 180,755 bubbles, which are polymorphic loci (see Fig. 1C), and 631,400 distinct alleles, which are possible sequences in each bubble or polymorphic locus (Fig. 1C). To ensure all genotypes were derived from reliably assembled sequences, we validated the assemblies with Flagger[16] and excluded the genotypes supported by collapsed, duplicated or low coverage regions (Supplementary Fig. 1, Methods). In the best assemblies, 98% of alleles were supported by valid regions, which is comparable to the HPRC assemblies (Fig. 1E). In a subset of 43 lower coverage assemblies, the number of alleles supported by valid regions was as low as 45%. However, Flagger may not be suitable to validate lower coverage assemblies where the sample size in each region is too small (Supplementary Fig. 2) because it needs to fit a mixture model on genome coverage data[16]. As expected, singleton alleles that were observed only once were the most likely to be called from unreliable sequences, with 42,881 of 215,578 singleton alleles (19.9%) being rejected by Flagger, while very common alleles were the least likely to be rejected (Supplementary Fig. 3). After excluding genotypes from invalid regions, we count a total of 178,188 reliable bubbles, involving 501,967 non-reference nodes and constituting 584,146 alleles. Some of these alleles were found in biallelic loci (150,942 alleles) but most were found in complex polymorphic loci (433,204 alleles) where we count up to 560 alleles in the same locus (Fig. 1F). As expected, the few loci with extremely large numbers of alleles are unstable simple and short tandem repeats, which naturally create many alleles but are difficult to align and require additional analysis (Supplementary Fig. 4). Of the 185,926 singleton SV alleles that were observed only once, 77.4% occur within 100 bp of a polymorphic locus with more than 2 alleles. Ancestry mapping based on SNVs (Supplementary Fig. 5) and pangenome alleles (Supplementary Figs. 6 and 7) showed that the majority of GA4K probands are of European-ancestry (EUR) and in line with self-reported ethnicity characteristics of the GA4K cohort[17] (Methods).

### Repeats and duplications are major contributors to structural variation

The non-reference nodes in the pangenome graph contributed approximately 610 Mbp of non-reference sequence, of which 184 Mbp was derived from HPRC genomes and 426 Mbp from GA4K genomes. We wanted to know what contributes to this increase in the size of the pangenome. RepeatMasker found that 74.2% of the content in non-reference nodes and that the leading contributors were simple repeats, satellites, L1s, and Alus (Fig. 2A). In terms of alleles, the pangenome contained 57,129 insertions, 50,011 deletions and an additional 418,302 variants with paths that pass through complex bubbles. These complex bubbles represent multiple deletions, insertions, or substitutions of DNA segments in loci such as STRs or VNTRs where structural variation from multiple genomes overlaps. Moreover, we discovered 1056 full-length LINE polymorphisms (Methods) and 15,598 full-length SINE polymorphisms (Fig. 2B and Supplementary Fig. 8), of which 340 LINEs and 3664 SINEs are unique to GA4K (Supplementary Fig. 9).

Next, we aligned the non-reference nodes to the CHM13v2 reference and found that 24.5% of all non-reference sequences (8.9% in HPRC and 15.6% in GA4K-only) were not repeats but mapped to some region in the genome (Fig. 2C), suggesting duplications or other rearrangement events. Overall, 98.7% of all non-reference sequences mapped to repeats or other parts of the genome, leaving 1.3% (7.5 Mbp) of the pangenome as putatively novel sequences. Novel sequences from the GA4K proband genomes accounted for 0.8% (4.7 Mbp) of the pangenome. Many of these novel sequences assigned to the pangenome were short, with only 21% being longer than 100 bp (Fig. 2C).

Finally, separate from the non-reference nodes that were included in the graph, there are also unanchored contigs in both HPRC and GA4K assemblies (Fig. 2D). We independently confirmed some of the

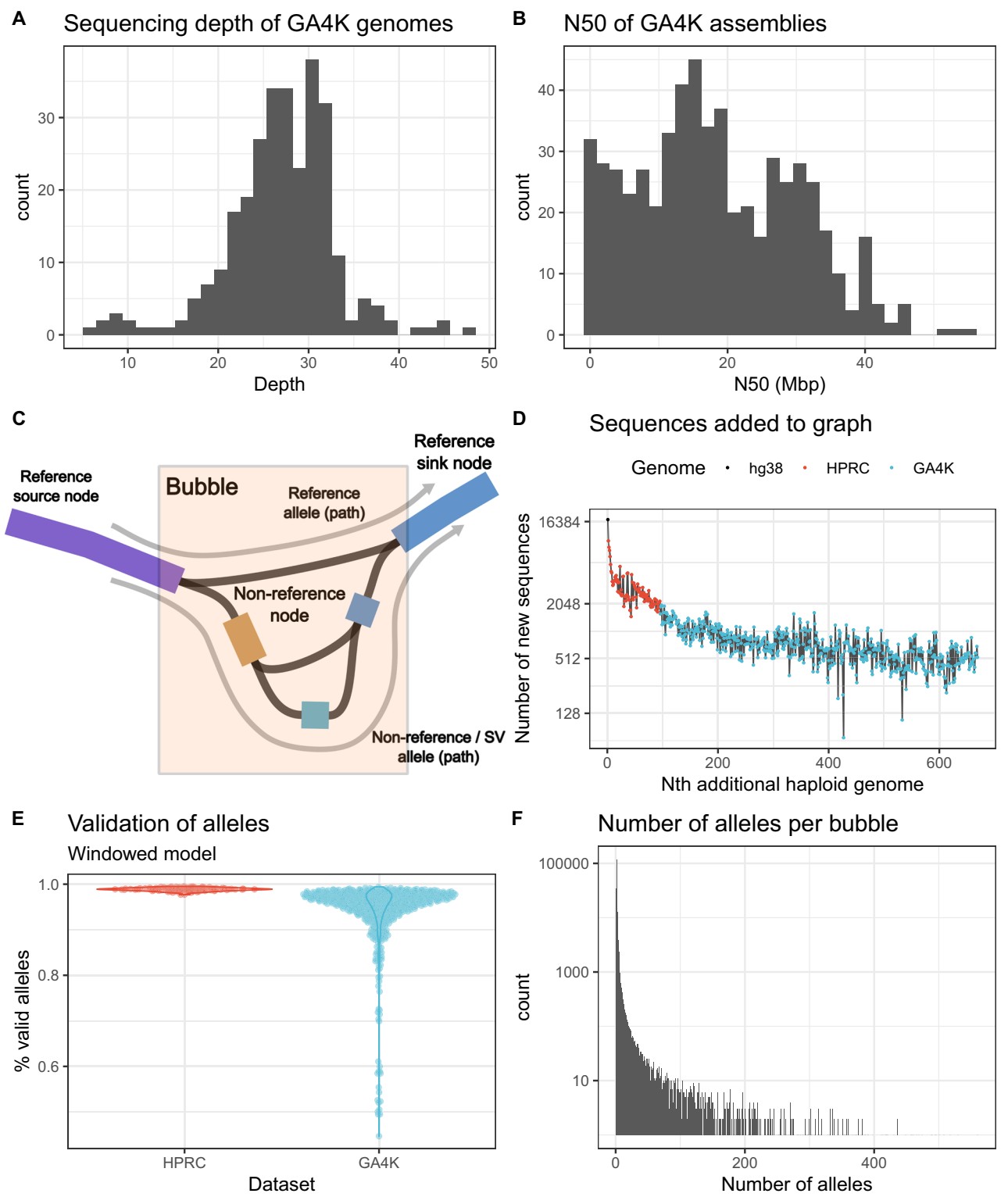

**Fig. 1 | Construction of the pangenome. A** Distribution of HiFi sequencing depth across the proband genomes. **B** Distribution of assembly contiguity (N50) across the proband diploid assemblies. **C** Representation of a polymorphic locus in a genome graph. A bubble begins at a source node when at least two genomes are different and ends at the sink node, where all genomes are the same. Paths from the source node to the sink node are alleles. Non-reference nodes are new sequences found in one of the 668 assemblies. **D** Growth of the pangenome as new genomes are added ($n = 668$). **E** Proportion of genotypes supported by sequences that pass validation with Flagger. **F** Distribution of the number of alleles observed in each bubble among the 668 genomes.

unanchored contigs for one offspring via coverage from mapped srWGS sequencing reads from that sample and its parents, and validate that contigs inherited from one parent (where there was high coverage from the offspring and only one parent) were unambiguously almost always inherited only from the appropriate parent, (Supplementary Table 1), indicating that these unplaced portions of the pangenome can be additional non-reference, family-inherited DNA sequence. We also confirm RNA transcription from these short-read WGS-validated

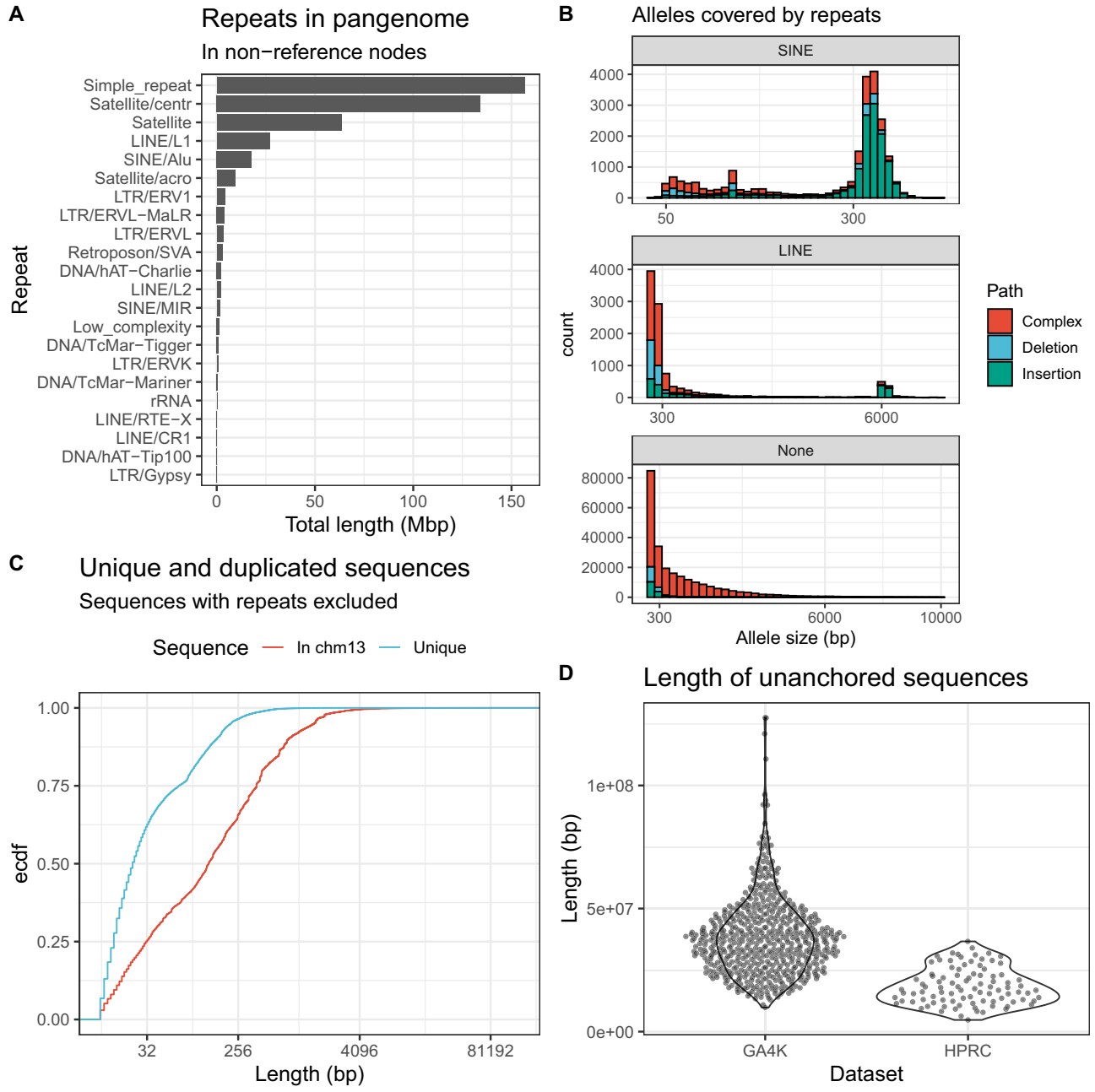

Fig. 2 | **Contents of the pangenome. A** Length of pangenome (non-reference graph nodes) that is masked with RepeatMasker. **B** Length distribution of alleles that are covered by SINE and LINE sequences and alleles that are not covered by repeats. **C** Length distribution of sequences that map to CHM13v2.0 but are not repeats and of sequences that do not map to CHM13v2.0 and are not repeats. **D** Total length of the sequence in each sample that is not anchored in the pangenome.

haplotype-resolved unanchored contigs by the mapping of phased reads from IsoSeq sequencing of the offspring (Supplementary Fig. 10), showing that these sequences could potentially contain active genes not currently captured in the reference genomes. In these unanchored contigs that are not in the graph, an average of 69 kbp of sequence per genome did not contain repeats and did not align to CHM13v2 (Supplementary Fig. 11).

**Calling SVs with a pangenome graph improves error rates**
We expect pangenome graphs to recover the SV alleles called by other long-read methods. We calculated the recall and precision of minigraph SVs (non-reference alleles in the genome graph) over the 287 probands against the SVs obtained with PBSV (Methods), which uses unassembled PacBio HiFi reads aligned to the GRCh38 reference genome. This yields a

two-dimensional distribution describing the recall (Supplementary Fig. 12A) and precision (Supplementary Fig. 12B) for each minigraph SV in each sample, which we visualize as a heatmap (Fig. 3A). We note that most SVs achieve very high precision and recall, while a small number show lower precision or lower recall. Overall genotypes, minigraph achieves a recall of 0.78 and a precision of 0.80 against PBSV, which is in line with previous benchmarking of these methods[16] in difficult regions of the genome. A similar sensitivity is also achieved when comparing to chromosomal microarray (CMA) results where minigraph recalls, on average, 79.5% (median 100%) of CMA SVs in each sample (Supplementary Fig. 13, Methods). Since no truth set is available, we could not directly evaluate the true positive rates of minigraph and PBSV. However, there is an identical twin pair with a shared phenotype in the GA4K cohort that we can use to explore the rate of SVs that replicate as a proxy

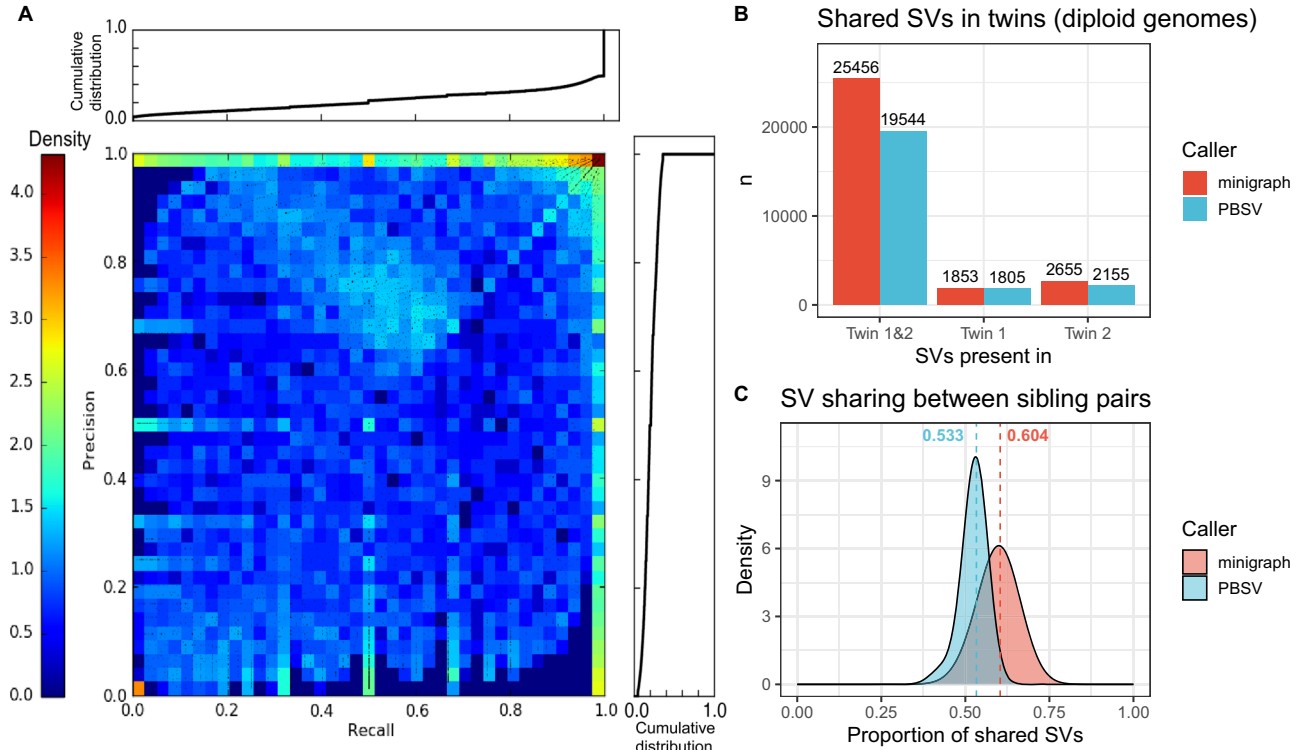

**Fig. 3 | Validation of SV calls. A** Recalls and precisions of minigraph SV call against PBSV shown as a 2D density distribution, with cumulative distributions in the margins. **B** Allele sharing between the GA4K twin pair, calculated with PBSV versus minigraph. **C** Allele sharing distribution between siblings in GA4K (twins and low coverage pairs excluded), calculated with PBSV versus minigraph.

for the true positive rate. We found that PBSV calls a total of 23,516 SVs, of which 19,547 (83.12%) are replicated in both twins (Fig. 3B). Meanwhile, minigraph calls 29,964 SVs, of which 25,456 are in both twins (84.96%). This boost in the number of detected SVs is corroborated by previous findings showing an increase in sensitivity over reference-based methods[16]. These results also indicate that false positive and negative rates of PBSV and minigraph heavily impact allele sharing since 15.04% (minigraph) and 16.88% (PBSV) of alleles were detected in only one of the twins. Thus, we consider increased allele sharing between siblings to be evidence of lower false positive and negative rates. We then explored allele sharing within the other 58 GA4K families in which at least two siblings were sequenced. In high-quality pairs where both siblings were sequenced at a depth above 20× minigraph shows an average of 7.1% more allele sharing than PBSV (Fig. 3C). As expected, lower coverage samples show less allele sharing due to their higher error rates (Supplementary Fig. 14). If we include pairs sequenced at a lower depth, siblings share on average 3.3% more alleles with minigraph than with PBSV (Supplementary Fig. 15). When randomly permuted sibling pairs, minigraph shows an average of 7.5% more allele sharing (Supplementary Fig. 16). While allele sharing indicates a lower overall error rate, it is affected by the different SV merging strategies employed by the two methods and cannot distinguish between false positive or false negative errors. Thus, we checked Mendelian violations in the GA4K232 trio in an attempt to disambiguate these two types of errors (Methods) and found that minigraph has a lower false positive rate and false negative rate relative to PBSV (Supplementary Table 2).

**Pangenome graphs reveal rare SV alleles in haplotype-resolved assemblies**
On average, we genotyped 18,326 non-reference SVs per haplotype (Supplementary Fig. 17A), or 28,261 SVs per diploid genome (Supplementary Fig. 17B), a figure that is in line with previous findings[6], but that is also influenced by assembly quality (Supplementary Fig. 18) and

genome diversity. At the same time, PBSV calls 22,428 SVs per diploid genome (Supplementary Fig. 17C). We wanted to characterize the population frequency of alleles in this dataset to identify rare variants in our set of individuals. To achieve this, we split alleles into three groups: those that are common to both cohorts, those that are unique to HPRC, and those that are unique to GA4K. As expected, the majority of alleles were observed in both cohorts, and their frequency distribution features the full range of rare, common, and nearly fixed alleles (Fig. 4A). More precisely, we observed 185,926 singleton alleles, 389,983 alleles with a frequency below 10% and 66,034 alleles with a frequency above 90%. In total, 314,981 alleles were observed in both datasets, 64,614 were unique to HPRC, and 204,551 were unique to GA4K. Also, 13,286 alleles that occur at <1% frequency in HPRC are, in fact, more common in GA4K, while 186,106 alleles that occur in GA4K at <1% frequency are not observed in HPRC. The most common allele unique to GA4K occurs in 88 out of 574 haplotypes (Fig. 4A bottom). Similarly, the most common allele unique to HPRC occurs in only 23 out of 90 haplotypes. As expected, the allele frequency distribution of alleles that are unique to GA4K and HPRC is heavily skewed toward rare variants that occur with a frequency below 10% (Fig. 4B). To ascertain how many singletons are due to sampling error, we genotyped the 88 haplotype resolved HGSVC assemblies[6] against our genome graph. As a result, we found that the HGSVC assemblies contain 17.0% of the HPRC singletons but only 4.85% of the GA4K singletons, indicating that the sampling error is smaller in the GA4K population.

Next, to begin exploring the properties of rare alleles, we categorized alleles into insertions, deletions, and complex events that may not be simple insertions or deletions relative to the reference and checked their size. We found that the average allele was 8.7 kbp long, with the expected peak at 300 bp (corresponding to Alu-related events), and that some alleles could reach up to 100 kbp in length (Fig. 4C). The very long alleles are created by expansions and

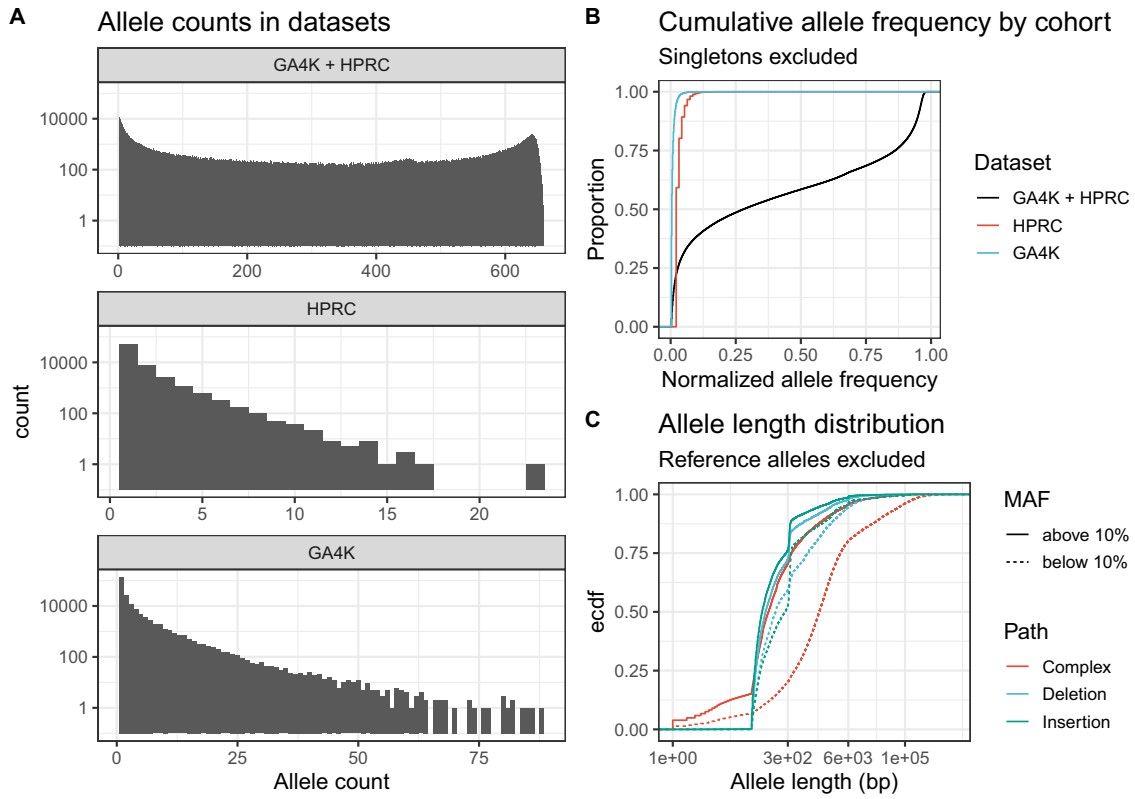

**Fig. 4 | Frequency of SVs. A** Frequency (count) distribution of alleles that are found in HPRC and GA4K, HPRC-only, and GA4K-only. **B** Cumulative distribution of allele frequencies (scaled by the size of HPRC and/or GA4K) in these three subsets of alleles. HPRC-only and GA4K-only variants are skewed towards rare frequencies. **C** Length distribution of non-reference SV alleles, stratified by frequency and type.

contractions of tandem repeats (Supplementary Fig. 19), inversions, or when large sequences are replaced by a much smaller sequence in one of the genomes. Notably, rare SVs were found to be longer than common SVs. Then, we assessed if complex, insertion or deletion GA4K-only SVs follow different frequency distributions (Supplementary Fig. 20). We found that complex SVs are the most skewed towards rare alleles, followed by insertions and then deletions.

### Rare SV alleles are distributed across the genome and found in genes of interest

Next, we were interested in SV alleles that may have functional relevance. To this end, we focused on the 204,551 alleles that were unique to GA4K, of which 132,391 SVs are singletons that were observed only once in GA4K. We observed that these alleles occur in hotspots of structural variation near telomeres and centromeres and that they sometimes overlap with genes and exons (Fig. 5A). We counted GA4K-only SV alleles in 1 Mbp windows and found 312 such SV hotspots in the top 10th percentile containing more than 171 SVs (Supplementary Fig. 21). Overall, we found 73,982 alleles within 7644 genes (9.68 alleles/gene) (Supplementary Fig. 22), of which 18,095 were within 3772 exons in 3112 genes (5.81 alleles/gene) (Fig. 5B). In particular, 1,383 alleles overlap 306 OMIM[18] exons in 275 OMIM genes (5.03 alleles/gene) that were previously associated with Mendelian diseases and phenotypes (Fig. 5C). Next, we checked if singleton alleles are enriched or depleted between intergenic, genic or exonic regions of the genome. When binning these alleles by frequency, the majority were singletons and rare variants. Singleton SVs accounted for 51,733 SVs in 6638 genes (7.79 alleles/gene), 13,083 SVs within 2932 exons in 2530 genes (5.17 alleles/gene), and 978 SVs within OMIM exons in 242 OMIM genes (4.04 alleles/gene). As expected, the frequency spectrum in exons and OMIM exons showed the strongest skew towards rare alleles, followed by intra-genic regions and intergenic regions (Fig. 5D).

In particular, 72.4% of SVs in exons and 70.7% of SVs in OMIM exons were singletons. In comparison, singleton SVs were slightly less represented in genic regions (69.2% of SVs were singletons) and much less represented in intergenic regions (62.7% of SVs were singletons). Of the GA4K singletons, 94,875 SVs overlap a gnomAD-SV interval[19]. Since gnomAD-SVs are not sequence resolved, this indicates that most singletons may either exist at a low frequency or occur in the same SV hotspots as in the broader population.

### Improved rare variant calling by joint graph and reference-based approaches

We have noted previously that rare HiFi-GS SV variants have higher parental transmission than rare short-read SV calls[20]. However, both pangenome and reference methods still display substantial false positive rates for rare SVs since every false positive will occur only a few times and be mistaken for a rare SV. We hypothesized that a consensus of reference-based and assembly-based methods would improve the precision of rare SVs over reference-based methods alone. First, we confirmed our expectations by benchmarking on the HG001 GIAB truth set (Supplementary Table 3), where the consensus SV set showed higher precision while being 17.1% smaller than minigraph and 11.1% smaller than PBSV. Then, to test if the proportion of false positives relative to true positives was reduced by such an approach in GA4K, we used an independent set of Illumina srGS Manta[21] SV calls (Methods) as the source of truth and investigated the precision of replicating common and rare (<5% MAF) SVs against this truth set. Reference-based SV calls alone are replicated with a precision of 38% across all frequencies, while reference-based rare SV calls are replicated with a precision of 11%. However, if we consider the consensus of reference-based PBSV calls and assembly-based minigraph calls, SVs of all frequencies are replicated with a precision of 56% and the rare SV subset is replicated with a precision of 61% (Supplementary Table 4),

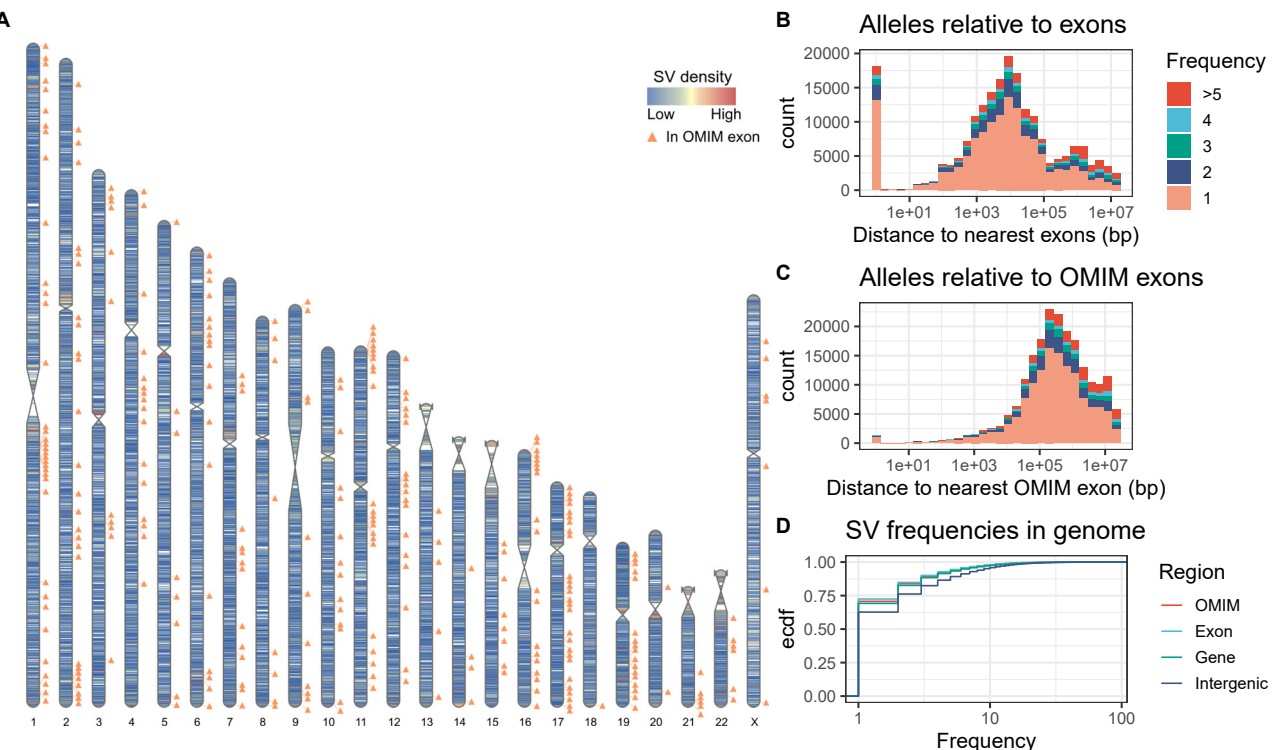

**Fig. 5 | Distribution of rare SVs in the genome. A** Density and hotspots of rare SVs that are found only in the GA4K cohort. The color scale denotes the number of rare GA4K-only alleles per megabase of the genome. OMIM exon positions are annotated. **B, C** Distance distribution of GA4K-only SVs relative to exons and OMIM exons stratified by frequency. **D** Allele frequency spectra of SVs in intergenic, genic, exonic, and OMIM regions of the genome.

indicating that combining assembly and reference-based methods improves the precision of rare SV calls significantly.

**Discovery of phenotypically impactful structural variants**

Finally, using this strategy, we curated the GA4K-specific minigraph alleles (not observed in HPRC) that are replicated by PBSV and potentially disrupt exonic sequences ($n = 924$). To focus on variants with potential phenotypic impact, we used the patient structured phenotype terms (HPO) to score candidate loci for each patient and limited to the top quartile of scores (phrank > 5)[22]. Among the rare SVs impacting the resulting 40 filtered exons, 10 were seen in highly polymorphic exons, suggesting that they are not evolutionary constrained or mapped to non-OMIM genes. In the remaining 30 exons, we observed 23 potentially pathogenic disruptions (2 previously reported pathogenic in ClinVar) in genes where loss-of-function (LOF) alleles are reported causes, but for autosomal recessive diseases (Supplementary Data 2). We checked for additional possible causal SNVs among the individuals with these 23 potentially deleterious alleles using DeepVariant[23] and did not find any. However, a subset of four alleles included a previously detected causal structural variant in *AARS2* (Supplementary Fig. 23), where the second variant is a likely pathogenic missense variant ([clinvar_ids: 213963], GRCh38 chr6:44311148 G > A, rs200105202 leading to c.595 C > T in NM_020745.4 and amino acid substitution p.Arg199Cys). In three other cases, the nature of the variant and its inheritance from the unaffected parent suggested low disease relevance. Also, a disease candidate inversion involving the *ACOX1* locus that rearranges several exons was observed. However, typically dominant *ACOX1* mutations are gain-of-function, and therefore, follow-up RNA expression studies are required (Supplementary Fig. 24). A paternally inherited rare deletion in *NLRP12* was observed with partial phenotypic fit, where variants have been reported to have variable penetrance. Finally, a novel diagnostic finding was uncovered in the maternal haplotype of one patient: a 14.5 kbp deletion in *KMT2E*

ranking in the top 5th percentile in phenotype fit score (phrank)[22] among all disease genes in this proband and was the highest scoring exonic rare SV affecting exons 9-13 in *KMT2E* (Fig. 6A–C). While the *KMT2E* variant is not exclusive to minigraph (and is validated by short-read WGS, Supplementary Fig. 25), it had not previously been prioritized for follow-up and clinical validation (see Supplementary Data 1 for testing history). This deletion is predicted to result in a premature stop and loss of function (NM_182931.3(*KMT2E*):c.729+113_1359-612del (p.Ala244*). The patient had a neurodevelopmental phenotype of hypotonia, macrocephaly, and developmental delay, overlapping the clinical picture described for *KMT2E* loss-of-function autosomal dominant variants. The maternal transmission was verified from sequence reads (Fig. 6D), the variant was validated by short-read genome data and was clinically confirmed by long-range PCR. Importantly, the mother had a history of cognitive delay and learning disabilities.

## Discussion

Pangenome graphs provide a comprehensive framework to study genetic variation and can explore complex loci that are difficult to characterize from pairwise comparisons to a reference genome. The ability of genome graphs to resolve recurrent structural variant biology[24] and repetitive DNA[25] was highlighted following the completion of the draft human pangenome reference[16]. In the case of rare genetic diseases, causal variants have a population frequency that is significantly lower than 1%. Expanding the collections of personal haploid assemblies in each population and generating deeper pangenome graphs will improve the filtering of common and rare SVs and help the identification of ultra-rare genetic variation in proband genomes.

Here, we relied on a progressive pangenome construction technique where each proband genome was added one at a time, revealing the SVs that are common and those that are less common. Such a

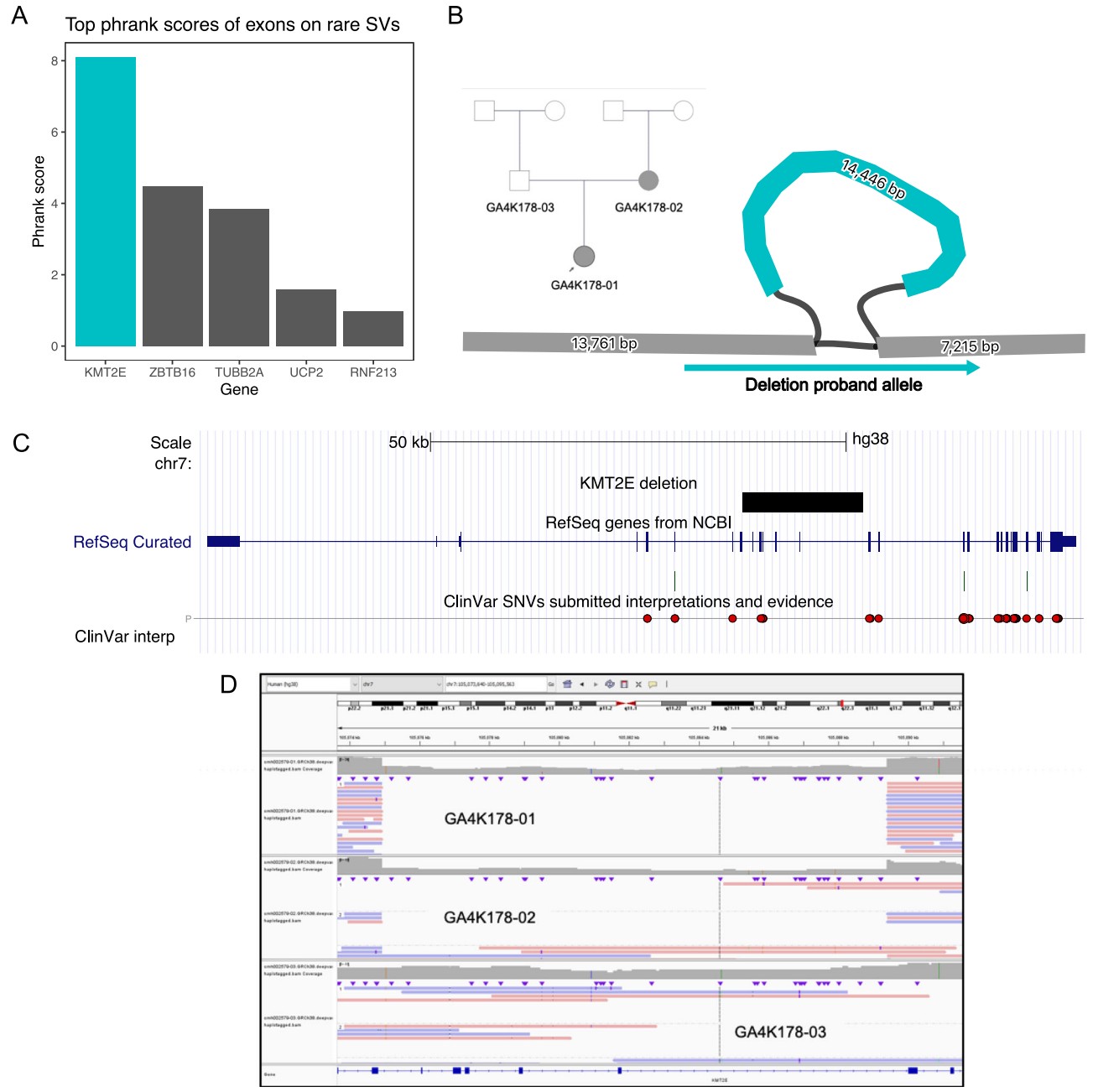

**Fig. 6 | KMT2E diagnostic deletion. A** Phrank ranking of rare SVs within exons. The *KMT2E* rare deletion ranks first among all rare SVs that overlap exons. **B** Pedigree of the 14,446 bp deletion and its representation in the genome graph. **C** UCSC Genome Browser view of the affected region. **D** Raw alignments of long-reads in the proband genome, the maternal and the paternal genomes, confirming the deletion, viewed in the Integrative Genomics Viewer (IGV).

progressive method is efficient, but it depends on the order of genomes that are incorporated and might miss events such as translocations[26]. An alternative could be a reference-free method such as the PanGenome Graph Builder[15], but further developments would be needed to implement adding genomes to an existing pangenome. Furthermore, this pangenome may be extended in the future by adding base-level variation with minigraph-cactus[27], which would reveal any small nested variation that may exist within structural variants and refine SV breakpoints.

Most of the additional pangenome content is composed of repeats, genomic duplications, and other rearrangements, which are difficult sequences that sometimes exceed even the length of long reads. This is especially the case for rarer SVs, which tend to be longer. These limitations increase the genotyping error rate, which

complicates the filtering and ranking of pathogenic SVs. Ensemble approaches that combine orthogonal approaches have been shown to improve variant calling[28–30]. Our analyses combined graph and reference-based approaches to improve the accuracy of SV identification.

The majority of structural variation in the pangenome continues to follow trends observed in previous assemblies, with the majority constituting repeat expansions and contractions. Despite this, it remains one of the largest collections of such rare disease genome assemblies to date, which we synthesized as a genome graph that organizes the structural variation in bubbles and allows queries using other assemblies. We released a useful resource to exclude common variation and keep SVs that are much more likely to be rare, even if the SV frequencies in this sample may not generalize to all populations.

Moreover, our resource allows users to expand this graph genome with their own assemblies, enabling rare SV discovery in any assembly. The pangenome graph and the process to iteratively add new assemblies are released and documented at https://doi.org/10.5281/zenodo.8309976. These resources will accelerate the interrogation of very rare SVs by the rare disease community, increasingly utilizing long-read sequencing as a rescue tool in unsolved diseases.

We established higher precision of rare SV consensus calls where reference and minigraph-based variants are concordant, which is particularly important for SVs outside known disease genes where the prior probability of pathogenicity is lower. High-quality rare SV catalog among undiagnosed cases will form the basis for new disease gene discovery.

We also applied phenotypically guided prioritization for manual curation involving only coding structural variation in known disease genes. Therefore, other novel disease genes and potentially non-coding variations can remain in our dataset and may be important for unsolved cases.

Moreover, our efforts to annotate the structural variation in these assemblies require an overlap with the existing functional annotation of the reference genomes. Better annotation of newly discovered regions would likely help identify more DNA with clinical significance and further increase diagnostic yield.

Finally, the SV alleles represented in the pangenome graph do not include sequences that could not be anchored to the pangenome. The estimated 69 kbp of unique sequences in contigs outside the pangenome were corroborated by family inheritance patterns and showed evidence of transcription. Placement of these contigs might require greater read lengths and sequencing depths, but further experiments would be needed to understand their potential function.

## Methods

### Ethical approval
The Institutional Review Board (IRB) of Children's Mercy Kansas City gave ethical approval for this work (Study#11120514).

### Age and sex reporting
Age and sex are reported for all study participants (Supplementary Data 3). Sex is assigned from self-reporting at enrollment and confirmed with genomic analysis. We focus all analysis on autosomes. Thus, the findings are applicable to both sexes, and no sex-specific analysis was performed.

### Trio assemblies with parental HiFi reads
To produce trio-binned assemblies, k-mer hash tables, including unique k-mers, were created for each parent from HiFi reads using yak count v0.1[5]. Proband HiFi reads were then assembled into two haplotypes with hifiasm v0.15[5] using the trio-binning method. The hifiasm k-mer frequency settings were adjusted to have a lower and upper bound of 1 to improve binning at lower parental coverage.

### Trio assemblies with parental short-reads
K-mer hash tables excluding unique k-mers were created for each parent from Illumina reads using yak count v0.1. Proband HiFi reads were then assembled into two haplotypes with hifiasm v0.15 using the trio binning method.

### Validation of diploid assemblies
We validated the diploid genome assembly using the Secphase–Flagger pipeline[16]. The HiFi reads were realigned to the combined diploid genome of each sample with minimap2[31]. The alignments were phased, corrected, and filtered using Secphase. For correcting alignments, we called biallelic SNVs with DeepVariant on the phased alignments. Then, we calculated the coverage across the diploid assembly with samtools depth. Finally, we clustered the regions of the diploid assembly into haploid, error, collapsed, and duplicated categories using Flagger. Specifically, we fitted the clustering model over 5 Mbp windows of the diploid assembly to account for local biases in sequencing depth.

### Ancestry mapping
We use somalier[32] to predict ancestry from the genotypes using principal components analysis based on 17,766 informative sites and 2504 reference samples from the 1000 Genomes Project.

### Creating genome graphs
We used minigraph-0.20 (r559)[13] with base-level alignments to build the genome graph with the command "minigraph -cxggs -t16 chm13v2.fa hg38.fa sample1.fa sample2.fa … sample668.fa > graph.gfa". We started with the CHM13v2[4] reference as a backbone and progressively augmented the graph with the hg38 reference, the HRPC genomes, and finally, the 574 GA4K haploid genomes. The order of the genomes to be added to the graph was determined lexicographically by sample name. To create a genome graph file suitable for publication, all stable sequence identifiers of each node were replaced with the sha256 hash of the GA4K assembly name concatenated to a random salt that is unique to each node. This yields a unique identifier for each stable sequence of each node from which the original assembly name cannot be reconstructed. This prevents linking nodes that were derived from the same assembly.

### Surveying additional sequences in the graph
We selected non-reference nodes that are above 100 bp in length from the genome graph. We ran RepeatMasker with the Dfam_2.0[33] database on the nodes to identify repeats. These nodes were also aligned to CHM13v2 with minimap -x sr to count how much sequence is not observed in the reference genome. Then we removed node intervals covered by RepeatMasker or CHM13v2 alignments with GenomicRanges::subtract to find unique sequences. Unique sequences shorter than 10 bp were ignored. We also spelled the path sequences and RepeatMasked full-length alleles.

### Calling genotypes
We called genotypes by realigning the assemblies back to the graph with minigraph -cxasm --call. The resulting genotypes were corrected by keeping calls derived from regions labeled as haploid. Calls derived from error, collapsed or duplicated regions were marked as invalid. We repeated the same with HPRC and HGSVC genomes. We merged the genotypes over the entire cohort by enumerating each observed traversal of a bubble as an allele in each sample. Then, we created a matrix where alleles are rows and columns are samples and where the presence of an allele is marked with 1 and its absence with 0.

### Annotating alleles
We categorize alleles according to the structure of the bubble in which they are found. Some alleles are simple paths that are clearly insertions or deletions, while some are complex paths that may be a combination of insertions, deletions, and alternate haplotypes relative to the reference. We ran RepeatMasker over the sequence of each allele. Full-length SINE and LINE were defined as alleles that are more than 80% covered by the repeat annotation and are 250–400 bp and 5000–10,000 bp in length, respectively. Each allele was overlapped with genes and exons using the lifted EBI GENCODEv38 r2[34] annotation that was published with the CHM13v2 genome[4]. Similarly, alleles were overlapped with OMIM exons using an annotation that was lifted over to CHM13v2. To overlap minigraph GA4K singleton SVs with gnomAD-SV, we lifted the gnomAD-SV annotation from hg19 to CHM13v2 and excluded BND calls.

## Population structure from SV genotypes

To compute the population structure of our datasets and the HPRC samples, we selected alleles that are called from a common set of regions that were assembled and passed quality checking in all samples. We augmented the HPRC genomes with 100 GA4K genomes of known EUR ancestry from previous ancestry mapping with somalier[32] in order to create a training set. Then, we used the R prcomp function followed by UMAP[35,36] on the first 45 PCs[36] on this training set to learn a projection, which we then applied to all HPRC and GA4K genomes.

## Comparison of SV genotypes from minigraph with SV calls by PBSV on GRCh38 and chromosome microarray

In order to evaluate the accuracy of our method, we compared minigraph calls to long-read PBSV PacBio calls. For this comparison, we restricted the minigraph calls to those with only two alleles and classified the less common variant as the ALT allele. We further omitted minigraph calls that are less than 100 bp away from regions missing in GRCh38. For this comparison, we considered both (1) the set of all minigraph calls remaining after filtering and (2) only minigraph calls whose ALT allele corresponds to a star (*) deletion. To compare the resulting call sets to PBSV, we then found all (>50 bp) PBSV calls within 100 bp of a minigraph call. We then quantified the correspondence between minigraph and PBSV by treating PBSV calls as the ground truth and calculating recall and precision accordingly:

$$R = \frac{\text{minigraph } calls\ with\ PBSV\ call}{\text{total } PBSV\ calls} \tag{1}$$

$$P = \frac{\text{minigraph } calls\ with\ PBSV\ call}{\text{total } minigraph\ calls} \tag{2}$$

When reporting the average values $R$ and $P$, we restricted to regions with at least one minigraph call and one PBSV call in at least one sample.

To compare against chromosome microarray (CMA), we lifted the hg38 CMA results to CHM13v2, filtered for assemblies with a minimum of 20× coverage, and reported the number of CMA intervals that overlap a minigraph non-reference SV allele.

When analyzing the shared genotyped SVs in pairs of siblings, we used SURVIVOR[9] to merge the non-reference PBSV calls of the siblings that are a minimum of 50 bp in size and at most 10% of the SV length apart. Similarly, we merged the non-reference minigraph alleles delimited by the same source and sink nodes of the SV bubble. Since the PBSV genotypes are not phased, we only considered the presence and absence of SV calls. The expected allele sharing is not necessarily 50% due to population structure among parents. To plot the allele sharing density, we adjusted the bandwidth parameter in the density kernel to smooth out the lower modes that are related to differences in sequencing coverage and to emphasize the highest modes that are related to PBSV and minigraph performance. We then repeated the analysis by randomly permuting sibling pairs 10 times and reporting the resulting distribution and its average.

We checked Mendelian violations on the GA4K232 trio, where parents and proband featured HiFi assemblies and PBSV calls. We used SURVIVOR as before to merge the PBSV trio genotypes. We defined Mendelian violations as genotypes that are impossible given the proband and parental genotypes. When an allele is homozygous in both parents, the proband must also be homozygous. If only one of the parents is homozygous, the proband must be at least heterozygous. Otherwise, we consider these events to be false negatives. Genotypes, where the proband has more copies of an allele than the parental genotypes allow, is a false positive events.

## Sensitivity of minigraph and PBSV consensus

To combine minigraph and PacBio PBSV calls into a single high-quality dataset and quantify the sensitivity of this set, we separated PacBio calls into those that match minigraph calls (the "concordant" set) and those that do not match minigraph calls (the "discordant" set) and then compared the rates at which the Illumina Manta calls[21] recall the concordant and discordant set. We used this to measure how the proportion of false positives changes relative to the proportion of true positives in the concordant and discordant sets. For this analysis, we focused on a set of 68 samples that have high-coverage PacBio sequencing data, high-coverage Illumina sequencing data, and minigraph calls. The reported numbers are averaged over these 68 samples. To create the concordant/discordant datasets, we first filtered the minigraph calls to regions consisting of exactly two alleles. We next created two sets of the minigraph/PBSV/Illumina calls—(1) sets containing all calls that pass quality filters and (2) sets containing only rare calls at <5% MAF. Then, for all and rare datasets separately, we found the PBSV calls that do/don't overlap a minigraph call to define the concordant/discordant datasets and then determined the fraction of these datasets whose calls overlap an Illumina Manta call.

## Unanchored assembly contigs

For all assemblies, we extracted contigs that do not align to the pangenome end to end. We aligned the sequences with minimap2 to CHM13v2.0 to find subsequences that align with the human genome. Then, we ran RepeatMasker to identify repeats in these contigs. We reported the number of unique sequences in each assembly as the number of base pairs that do not align to the pangenome, do not align to CHM13v2.0, and are not covered by RepeatMasker annotations.

For two trios where corresponding srWGS data was available, we used the assembled contigs for each of the two-phased haplotypes in the child as a personal reference genome and used DRAGEN to align the srWGS from each of the parents and from the child itself to the child's personal reference genomes. Examining the coverage in the assembled contigs that were completely unanchored by the minigraph at 500 bp bin resolution using most depth, we classified the bins as either being covered (>8× for the child or father of GA4K86-01, >3× for other parents), otherwise classified as uncovered. While the majority of bins in the unanchored regions are covered by all three members of the trio, there is a distinctive subset of over 1 Mb of sequence that is covered in the child and the expected parent for the phased haplotype for the personal reference, indicating there is still unique, inherited genomic sequence yet to be explored (Supplementary Table 1).

For one of these trios, in addition to the srWGS data, we also have Iso-Seq RNA expression for three different cell types (blood, iPSC, neuronally-differentiated iPSC), which we aligned to the paternal and maternal personal genomes of the proband. Barcoded primers were removed with PacBio's demultiplexing tool *lima*, followed by *isoseq refine* to assemble full-length non-chimeric transcripts (FLNC). Assembled FLNC transcripts were aligned to the hifiasm hap1 and hap2 assemblies of the proband using gapped minimap2. Mosdepth[37] was used to determine binned (500 bp tiles) coverage across the contigs for each of the assemblies.

The majority of the Iso-Seq expression was seen in the bins that are covered by both parents (as that is the largest proportion of the bins); however, among the bins that were uniquely covered by only one of the parents, we see Iso-Seq signal aligned to the paternally inherited genome only showed expression in bins covered by the father, and likewise Iso-Seq signal aligned to the maternally inherited genome only showed expression in bins covered by the mother (Supplementary Fig. 6).

## Reporting summary

Further information on research design is available in the Nature Portfolio Reporting Summary linked to this article.

## Data availability

The 5-base HiFi-GS, HiFi long-read transcript sequencing (IsoSeq), and WGBS raw and processed data, including assemblies and genotypes generated in this study, have been deposited in the dbGAP (https://www.ncbi.nlm.nih.gov/gap/) database under accession code phs002206.v4.p1. Raw and processed data are available under restricted access due to IRB regulations and informed consent limiting access to users studying genetic diseases. Data access is provided by dbGAP (https://dbgap.ncbi.nlm.nih.gov/aa/wga.cgi?page=login) for certified investigators with local IRB approval in place. The CHM13v2.0 reference genome is available for download at https://s3-us-west-2.amazonaws.com/human-pangenomics/T2T/CHM13/assemblies/analysis_set/chm13v2.0.fa.gz, and the GRCh38 reference genome is available for download at https://hgdownload.soe.ucsc.edu/goldenPath/hg38/chromosomes/. The GA4K genome graph, allele definitions, and their frequencies, together with related data on assembly size, read depth, and validation with Flagger and Repeat-Masker results, are available for download at https://doi.org/10.5281/zenodo.8309976.

## Code availability

Custom scripts can be downloaded at https://doi.org/10.5281/zenodo.8309976. These scripts were used to build the genome graph, genotype the assemblies, validate and merge the genotype information across samples, calculate SV lengths, and repeat mask SVs.

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

## Acknowledgements

We would like to thank all families for participating in the Genomic Answers for Kids study. This work was made possible by generous gifts to the Children's Mercy Research Institute and Genomic Answers for Kids program at Children's Mercy Kansas City. We also would like to thank Nick Nolte, Dan Louiselle, and Rebecca Biswell for their work in sample processing, Laura Puckett and Adam Walters for their work in

library preparation and sequencing, and the clinical coordination team led by Bradley Belden for their work in clinical coordination. We also would like to thank PacBio for sequencing support for a subset of the samples. T.P. holds the Dee Lyons/Missouri Endowed Chair in Pediatric Genomic Medicine, and E.G., holds the Roberta D. Harding & William F. Bradley, Jr. Endowed Chair in Genomic Research. C.G. is supported by the NSERC PGS D award. G.B. is supported by a Canada Research Chair Tier 1 award and an FRQ-S Distinguished Research Scholar award. The Canadian Center for Computational Genomics (C3G) was supported by a Genome Canada Genome Technology Platform grant. This research was enabled in part by support provided by Calcul Quebec and the Digital Research Alliance of Canada.

## Author contributions

G.B. and T.P. conceived and designed the study; C.G. and T.C. contributed to study design; E.G.F., I.T., W.B.R., and G.E. prepared, provided and/or analyzed clinical samples and associated data; C.G., C.S-S., W.A.C., and J.L. provided bioinformatics support; C.G., C.S-S., W.A.C., G.B., and T.P. analyzed the data and interpreted results of experiments; C.G. prepared figures and drafted paper; G.B. and T.P. edited and revised paper; all authors approved the final version of paper.

## Competing interests

Juniper Lake is a current or past employee of Pacific Biosciences. The remaining authors declare no competing interests.
