## [Peer Review File · Nature Communications]

Pangenome graphs improve the analysis of structural variants in rare genetic diseasesREVIEWER COMMENTS

Reviewer #1 (Remarks to the Author):

The authors present an internally developed graph genome workflow, based on HPRC data, and employ this workflow to study GA4K data. Their analysis resulted in the discovery of a diversity of genomic variation. The manuscript is well written, however the workflow appears ad-hoc, is not reproducible, and is not benchmarked. In my opinion, the study possesses potential. However, the authors might want to either concentrate on diagnostic yield with longread GS, or develop the study into a comprehensive bioinformatic article. Currently, it feels like they are aiming for both those aspects and not fully achieving in either.

Major comments:

1. The authors have not provided any tool to reproduce the genome graph. I would suggest making the software available on a website such as git hub. Overall, their graph workflow appears to be ad-hoc, consisting mainly of the already published minigraph.

2. The majority of their genomic findings are not novel, it is for instance well known that the majority of SV constitute repeat expansions/retractions.

3. The genome graph has not been compared against existing software, making it difficult to assess its true value.

Minor comments.

4. More detailed information about the cohort would be beneficial. A table outlining the specifics of what genomic analysis (GS, ES, CMA...) was conducted and how the data was analyzed (SNV, SV, assembly used) before the current study is recommended.

5. The authors need to clarify the sentence below, do they mean that for 90% of the cohort clinical genomes were done and only analyzed for SNVs.

“By design, this cohort was enriched for difficult to solve cases and more than 90% of the probands were undiagnosed even after standard clinical sequencing and exploration of putatively causative single nucleotide variants (SNVs).”

6. In the results section please provide basic HiFi sequencing parameters such a read depth (mean, median and range) and N50.
7. The authors should explain the terms "bubble" and "distinct allele" in the sentence "Using the resulting graph we genotyped the assemblies and observed 180,755 bubbles and 631,400 distinct alleles"
8. Please provide more context about the twin pair in the GA4K cohort including the type twins and if they shared phenotypic presentation.
9. The total number of identified hotspots of structural variation should be specified as well a definition for how the authors define such a hotspot.
- 10.. The total number of genes and exons involved, as well as the per gene frequency of new sequence insertions should be specified when stating: "Overall, we found 73,982 alleles within genes (Fig S14), of which 18,095 were within exons."
11. Please define the term "singleton" when referring to SVs.
12. The sentence, "In comparison, singleton SVs were slightly less common in genic regions (69.2%) and much less common in intergenic regions (62.7%)." is difficult to understand and requires further clarification.
13. A comparison of the KMT2E deletion to SV calling from short-read GS with multiple callers is necessary. Also, it would be helpful to include the methods used prior to HiFi GS and how the data was analyzed. Please incorporate a screenshot from IGV of srGS data in figure 6.

Reviewer #2 (Remarks to the Author):

The authors investigate using pan genome graphs for rare disease analysis with a particular interest in SVs. The idea is quite interesting since the allele frequency spectrum that applies to rare disease seems to be far below what has been possible with most pangenomes to date. They did haploid assembly on 287 proband, and with 94 additional haploid assemblies they created a pangenome. From that they

found SVs. Every additional genome they added provided 500 additional sequences, which, as they point out, means that many more alleles are to be discovered. It also means that this data set will not be great for determining frequencies for alleles that are on the rarer end of the spectrum, which is directly at odds with the author's objective.

My main issue with this paper is the focus on rare SVs and the small sample size. I think they could (but don't) make the case that even with the small sample size, if performance improves on the relatively rare SVs then we should expect that to continue to be true for truly rare variants. But they never make that case. The one section of the paper that addresses improving rare SV calling is also confusing. There is this hypothesis "We hypothesized that a consensus of reference-based and assembly-based methods would improve the precision of rare SVs over reference-based methods alone." I do not really see how the following experiment tests this hypothesis.

I am troubled by the statement "As expected, singleton alleles were the most likely to be called from misassembled sequences, while very common alleles were the least likely" Rare disease research is about singleton alleles. Since the signal we care the most about is also the most troubling, I would like the authors to expand on this point. How many singletons were found? How many were called from misassembled sequences? How do we know they are misassembled? I want to be reassured that we are not throwing away important sources of variation. I think right now they are just excluded.

"we count up to 560 alleles in the same locus" Wow! There are only 668 haplotypes. Please comment on what this locus is and what is known about it.

I liked the experiment that used the twins to determine false positive/negative rates. The language needs to be cleared up here because you cannot differentiate between a false positive and a false negative, which prevents conclusions around true positive rate etc. Specifically, I need some clearer mathematical justification that statements such as "Thus, we consider increased allele sharing between siblings to be evidence of lower false positive and negative rates." and "The increase in allele sharing between siblings suggests that the SV calls obtained with minigraph have a lower false positive and false negative rate." are true. I would expect that some of these SVs could be genotyped in the parents, so the authors may want to look at mendelian violations to help determine FP and FN.

In general, it would be nice if the discussion about the SV landscape broke the analysis up in the SV types. I would expect the numbers to be better for DEL and worse for INS, and better for big things and worse for little things. Summing things up into just one number is a little too high-level.

Given that assembly quality is such an issue, I would suggest moving those figures from the supplement to the main paper. This is one of the most important results of this paper.

Reviewer #3 (Remarks to the Author):

Summary

The manuscript “Pangenome graphs improve the analysis of rare genetic diseases” describes a study investigating a large cohort of 287 trios with rare pediatric disease from the Genomic Answers for Kids (GA4K) program. These probands from GA4K had undergone PacBio HIFI sequencing, and the authors took a unique approach for analysis of this dataset by building graph genomes for each case. They compared structural variant (SV) yield from the graph genomes to more traditional reference-based methods. The authors also incorporate 94 control haploid genomes released by the Human Pangenome Reference Consortium (HPRC) to better filter for rare variants. Examination of the functional SVs discovered from the graph found over 1,000 rare exon disrupting events. They further narrow this list to 30 SVs with potential disease relevance including a pathogenic deletion in KMT2E. Overall, I found the approach taken in this paper to be intriguing and certainly think the results are of interest to the field. The introduction does a fantastic job explaining why graph genomes could be of interest to the clinical genetic community and the benchmarking performed is mostly adequate. My greatest area of concern is the rare disease analyses, which overstates the utility of the graph method since the authors fail to perform an adequate comparison of pathogenic yield with both short read and ref-based long read methods. Based on the presented results it doesn't seem graph genomes actually improve the analysis of rare disease since the pathogenic variants found were also discovered in short reads. I don't think this is a huge problem as the utility of genome graphs is likely to come in interpretation of difficult to map regions, which currently cannot be easily interpreted. I have the following comments to help strengthen the manuscript.

Major Comments

1. There needs to be a clearer description of what prior genetic results went into filtering this cohort since that will have a dramatic impact on the expected diagnostic rate. It seems clinical sequencing was performed but is that whole genome or whole exome? Were only cases with pathogenic SNVs were removed but what about SV. Given SV are the focus of this paper it seems important to distinguish whether those have been investigated. The overall rare disease yield found in the study low so I am guessing additional SV filtering must have been applied.
2. The Iso-Seq analysis could be really interesting but is poorly described in the main text. Please add more detail such as the number of sites investigated and across how many samples. Also, what tissue was investigated and does the RNA seems to be coding or noncoding?
3. Benchmarking with long read data is challenging given a lack of truth data for hard to map regions. In cleaner non-repetitive genomic regions though short read WGS should be adequate for benchmarking. The authors perform this benchmarking but after already describing much dataset. I recommend incorporating this earlier in the benchmarking section of the results.

4. Assessing shared variants across family members can provide a rough measure of genotype accuracy but is still fraught with issues. For instance, a false positive SV that is an assembly error across many samples would be more likely to look shared between sibs even though false. A permutation approach building a distribution from random sib pairs could help protect against some of these technical biases and provide a better comparison between PBSV and the graph genome approach.

5. The authors clearly demonstrate the ethnicity of the HPRC are widely different from the GA4K, which would influence rare variant filtering. Given the majority of rare pathogenic variants fall in clean genic regions could the author not use the short read SVs from large-population cohorts (e.g, gnomAD) to further filter the GA4K during disease assessment.

6. The title of the paper is “Pangenome graphs improve the analysis of rare genetic diseases”, but the current analysis doesn’t do a proper assessment to allow for this statement. The lack of comparison between graph and ref-based methods (both long and short read) for the diagnostic analysis is glaring. It is really important to show whether the graph-based method adds any additional yield over the more traditional methods.

7. This study serves as a proof of concept of the power of graph genomes but until we have a better functional annotation of newly discovered regions these methods will not be fully appreciated. I feel this is needs to be mentioned in the discussion.

8. The abstract and title should be revised to better reflect the actual result of the study which shows minimal additional diagnostic utility under current clinical guidelines.

Minor Comments

1. I am confused by this sentence “We independently confirmed some of the unanchored contigs for one offspring via coverage from mapped srWGS sequencing reads from that sample and its parents, and validate that contigs inherited from one parent (where there was high coverage from the offspring and only one parent) were unambiguously almost always inherited only from the appropriate parent, (Table S1), indicating that these unplaced portions of the pangenome graph can be additional, potentially de novo, family-inherited DNA sequence.”

I don’t think anything can be said about de novo mutation here. Of course, any variant type can be de novo but seems very strange to highlight here given that is not really being looked at since the parents don’t have long read sequencing.

3. Heat map seems bimodal but hard to compare with heat map. Replacing or supplementing with something like a violin plot could better show the distribution of the precision and recall.

4. I would like to see per sample variant counts for PBSV and more detailed description of that callset given its importance to the study.

5. Which specific OMIM genes were used in the genes of interest analysis? A focus on autosomal dominant may help better understand false positive rate since we wouldn't expect very many disruptions of those.

6. The ACOX1 inversion doesn't appear to be exon disrupting and therefore likely has no predicted functional consequence.

7. Figure legends S15 and S16 are messed up

8. I don't understand the definition of "highly polymorphic (non-constrained) exons". Is the exon disrupted by many SVs so you don't think it is evolutionary constrained? Is this some sort of measure from gnomAD?

9. Fail to see why this method may miss translocations. Please describe in more detail in the discussion.

Reviewer #4 (Remarks to the Author):

In this study, Groza et al use long-read (HiFi) sequencing on probands in 287 parent-offspring trios (Illumina on the parents). A pangenome graph was created on the assemblies of the probands along with the HPRC genomes. The pangenome graph was used to compare variants across individuals. A significant finding is transcripts mapping back to nonreference unanchored contigs. An expected finding is that combining minigraph+pbsv calls increases precision.

The major comment is that it was never quite clear just what the benefit was of using the pangenome graph. For example, it does not appear that an effort was made to create a unified callset using modern

approaches for merging calls (e.g. Jasmine <https://www.nature.com/articles/s41592-022-01753-3>). Furthermore, minigraph is in fact less accurate than simply calling variants using assembled sequences and dipcall, particularly for determining breakpoints.

Finally, a lot of researchers have been hoping that long read sequencing will help discover new causal variants. This study has the potential to answer how pangenomes increase diagnostic yield, but instead a high level approach was taken to quantify an increase in precision using graph+pbsv. The effect on diagnostic yield should be noted, first by detecting pathogenic variants in the short read data either with Manta, or a method that is much more precise (Delly, lumpy, and wham all have greater precision, which despite the drop in sensitivity, are more useful for detecting pathogenic variants than Manta). Then a comparison with a single method (such as pbsv, or better dipcall on the assembled sequences).

The rest of the comments are minor.

Methods: the HPRC used a combination of minigraph and cactus to refine breakpoints around structural variants, this may be tried (or noted).

How many of the rare SVs are near tangles (VNTR sequences) in the pangenome graph?

Minor: the very long novel alleles (>50kb) are worth more consideration. First, try remapping the novel allele + some flanking sequence to CHM13 T2T (or just hg38) to make sure these are not artifacts of minigraph alignment. Next, it would be good to know which genes map to these novel sequences.

Minor: enrichment (or depletion) analysis should be done for singleton SVs in genes/exons/OMIM exons.

Minor: what is the percent reduction in calls when combining minigraph + pbsv?

Minor: generally, the calls based on de novo assemblies (using dipcall) are superior to pbsv calls. For the assemblies that are high quality (>30x input coverage), it would be good to redo some of the joint calling analysis using dipcall variants instead of pbsv. Furthermore, it's unclear whether the benefit was inherent to use of the graph itself, or simply a separate variant calling approach.

Minor: The low mapping rate of non-reference nodes is puzzling, even if they are not annotated as repeat nodes. In the past, I've tried to see what insertion sequences (e.g. equivalent to non-reference nodes) have no match in the genome using: 1. repeat masking of sequences, then mapping back to the genome, then mapping back to nr. Almost everything had some decent alignment.

Minor: it would be good to start with a figure describing the dataset: distribution of coverage, assembly quality, etc. This will ground the readers in the quality of the dataset.

Use uniform processing of the HPRC assemblies to calculate the number of SV in OMIM genes. The HGSC assemblies may be added for improved sampling of low allele frequency.

REVIEWER COMMENTS

Reviewer #1 (Remarks to the Author):

The authors present an internally developed graph genome workflow, based on HPRC data, and employ this workflow to study GA4K data. Their analysis resulted in the discovery of a diversity of genomic variation. The manuscript is well written, however the workflow appears ad-hoc, is not reproducible, and is not benchmarked. In my opinion, the study possesses potential. However, the authors might want to either concentrate on diagnostic yield with longread GS, or develop the study into a comprehensive bioinformatic article. Currently, it feels like they are aiming for both those aspects and not fully achieving in either.

We understand that the main objectives of our study were not sufficiently clear in the first version of the manuscript. This was raised by multiple reviewers. We have now addressed this in detail in our point-by-point responses below.

Major comments: 1. The authors have not provided any tool to reproduce the genome graph. I would suggest making the software available on a website such as git hub. Overall, their graph workflow appears to be ad-hoc, consisting mainly of the already published minigraph.

The reviewer is correct in that we do not develop a novel method for creating genome graphs; the existing methods perform well and have been extensively benchmarked by the Human Pangenome Reference Consortium. Instead, we are applying an existing method to the largest collection of rare disease HiFi assemblies in an effort to characterize structural variation in this cohort. We now clarify in the Methods that we used the standard unmodified minigraph workflow to generate the pangenome graph, and we have added specific version number and command line options used for full transparency.

“We used minigraph-0.20 (r559) (Li, Feng, and Chu 2020) with base-level alignments to build the genome graph with the command “minigraph -cxggs -t16 chm13v2.fa hg38.fa sample1.fa sample2.fa ... sample668.fa > graph.gfa”.”

We also agree with the reviewer that our workflow should be reproducible. While the assemblies that we've generated contain identifiable information that cannot be released without control-access, we are now publicly releasing the genome graph and observed alleles of the cohort, which will allow others to genotype their own assemblies against this GA4K collection and enable their own filtering for rare SV alleles. The released data may be found at <https://doi.org/10.5281/zenodo.8309976>. Included in this release is a document outlining our workflow, from graph creation with minigraph, genotyping assemblies against the graph, merging SV genotypes from multiple samples, and counting alleles. We added a new “Availability of data and materials” section to the manuscript:

“The GA4K genome graph, allele definitions and their frequencies, together with related data on assembly size, read depth, validation with Flagger and RepeatMasker results are available for download at <https://doi.org/10.5281/zenodo.8309976>.”

We also added sentences to abstract and discussion reflecting this high utility, unparalleled resource for the community.

“Our deep pangenome and iterative process for harmonized interrogation of additional assemblies are made accessible to the community facilitating discovery of new SVs within allele frequency spectrum relevant to genetic diseases.”

“The pangenome graph and the process to iteratively add new assemblies is released and documented in <https://doi.org/10.5281/zenodo.8309976>. These resources will accelerate interrogation of very rare SVs by the rare disease community increasingly utilizing long-read sequencing as a rescue tool in unsolved disease.”

2. The majority of their genomic findings are not novel, it is for instance well known that the majority of SV constitute repeat expansions/retractions.

While some of our results are expected given previous results in human genomics, we believe our analyses are a notable contribution since we are doing them in a novel context of 668 genome assemblies. For example, while we know most structural variation would consist of repetitive sequence, we don't know the exact ratio of repeat to unique sequence in a pangenome of this size. By annotating repeats in the pangenome, we found that a remaining 1.3% of the extra pangenome growth is unique sequence. Moreover, while common variation is often studied at the assembly level, reliable evaluation of rare variation that occurs at <1% frequency has never been done before, especially in the context of rare genetic disease. Here, we made use of the minigraph concepts that frame structural variation as bubbles in which alleles are unique paths to reliably match and count alleles across many assemblies, including in complex loci that are not representable by the VCF format. Moreover, the minigraph approach lets us handle nested structural variation, that is, variants that occur within other non-reference SV alleles that are not represented by CHM13v2 or hg38. This is particularly important, since as we found in this study, most structural variation occurs in multi-allelic loci. To address these points, we added the following to the discussion:

“The majority of structural variation in the pangenome continues to follow trends observed in previous assemblies, with the majority constituting repeat expansions and contractions. Despite this, it remains one of the largest collections of such rare disease genome assemblies to date, which we synthesized as a genome graph that organizes the structural variation in bubbles and allows queries using other assemblies. This resource can be used to exclude common variation and keep SVs that are more likely to be rare, even if the SV frequencies in this sample may not generalize to greater populations.”

3. The genome graph has not been compared against existing software, making it difficult to assess its true value.

We agree that new software must be benchmarked against existing solutions to assess its true value. Here, as mentioned above, we used minigraph, which is an existing tool whose utility was already demonstrated to build the first draft human pangenome reference on 90 diverse genomes (Liao et al. 2023). In those findings, minigraph was shown to have similar recall and precision when genotyping structural variation as other structural variant callers. We then explored the practical research application of taking the HPRC resource and applying it to investigate a research cohort, demonstrating the feasibility of assembling a pangenome with 668 haplotypes using minigraph and describing the resulting pangenome graph. For example, we found that 13,286 SVs that occur at <1% frequency in HPRC are in fact more common in GA4K. At the same time, 186,106 SV alleles that occur in GA4K at <1% frequency were not observed in HPRC. By combining existing resources in this manner, we are able to characterize structural variation in this research cohort of genomes. We added these points to the results section:

“We then created a pangenome graph with minigraph (Li, Feng, and Chu 2020), **which was previously tested and found to highly agree with reference-based methods** (Liao et al. 2023), to identify structural variants in the combined set of 668 haploid genomes together with two standard reference genomes (GRCh38 and CHM13v2).”

“Also, 13,286 alleles that occur at <1% frequency in HPRC are in fact more common in GA4K while 186,106 alleles that occur in GA4K at <1% frequency are not observed in HPRC.”

Minor comments. 4. More detailed information about the cohort would be beneficial. A table outlining the specifics of what genomic analysis (GS, ES, CMA...) was conducted and how the data was analyzed (SNV, SV, assembly used) before the current study is recommended.

We agree that listing the previous genomic analysis for each proband would highlight that this cohort enriched in difficult to diagnose cases. We have now added a supplementary table (Table S1) describing the previous assays that were previously performed on each proband and point to it in the results:

“By design, this cohort was enriched for difficult to solve cases and more than 90% of the probands were undiagnosed even after standard clinical sequencing and exploration of putatively causative single nucleotide variants and in many cases dedicated clinical microarray studies for structural variation (**Table S1**).”

5. The authors need to clarify the sentence below, do they mean that for 90% of the cohort clinical genomes were done and only analyzed for SNVs. “By design, this cohort was enriched for difficult to solve cases and more than 90% of the probands were undiagnosed even after

standard clinical sequencing and exploration of putatively causative single nucleotide variants (SNVs).”

The GA4K cohort centers around the study of unsolved pediatric cases, with more than 90% of the probands in the cohort being individuals who underwent standard clinical sequencing and exploration of causative single nucleotide variants but a diagnosis was not found using these traditional methods. During this study, clinical or research molecular assays revealed a molecular diagnosis for 10% of cases, yielding 90% undiagnosed probands. We clarified the above sentence in the following way:

“In this cohort, more than 90% of the probands **remained** undiagnosed even after standard clinical sequencing and exploration of putatively causative single nucleotide variants (**Table S1**), with only less than 10% being eventually diagnosed. Thus, **this set of genomes is enriched for difficult to solve cases.** “

6. In the results section please provide basic HiFi sequencing parameters such a read depth (mean, median and range) and N50.

We have added information about these basic parameters in the main text and in the following new figure panels (**new Fig 1A-B**).

Fig 1: A) Distribution of HiFi sequencing depth across the proband genomes. B) Distribution of assembly contiguity (N50) across the proband diploid assemblies.

Moreover, we have added the following text to the result section:

“at a mean depth of 27X (Fig 1A, median 27X, range 6X-48X)”

“obtaining a mean N50 of 18.2 Mbp (Fig 1B, median 16.4 Mbp, range 78.6 kbp-55.3 Mbp)”

7. The authors should explain the terms "bubble" and "distinct allele" in the sentence "Using the resulting graph we genotyped the assemblies and observed 180,755 bubbles and 631,400 distinct alleles"

Indeed, these are new terms for most readers. We edited the text to mention a brief explanation for each term in the result section.

"Using the resulting graph we genotyped the assemblies and observed 180,755 bubbles, **which are polymorphic loci** (see **Fig 1C**), and 631,400 distinct alleles, **which are possible sequences in each bubble or polymorphic locus** (**Fig 1C**)."

We also refer readers to the illustration in Fig 1C that shows a visual explanation of the terms bubble and allele.

8. Please provide more context about the twin pair in the GA4K cohort including the type twins and if they shared phenotypic presentation.

The twins were identical and shared key features of a rare unsolved hematologic abnormalities, angioedema and susceptibility to infection. However, for public release, we only have permission to share: "identical twins with shared rare phenotype":

"However, there is an **identical twin pair with shared phenotype** in the GA4K cohort that we can use to explore the rate of SVs that replicate as a proxy for the true positive rate."

9. The total number of identified hotspots of structural variation should be specified as well a definition for how the authors define such a hotspot.

The reviewer raises a good question about the exact number of SV hotspots which we didn't define in the original version of the manuscript. Following this comment, we now binned the genome in 1 Mbp bins and counted the number of SV alleles per bin that are unique to the GA4K cohort. Based on the resulting distribution (**new Fig S21**), we defined the top 10% of bins with the highest number of SVs to be hotspots. In total, we found 312 hotspots that contain more than 171 SV alleles.

Fig S21: Density of GA4K-only SVs in 1 Mbp windows across the genome.

We added the following new text:

“We counted GA4K-only SV alleles in 1 Mbp windows and found 312 such SV hotspots in the top 10th percentile containing more than 171 SVs (Fig S21).”

10.. The total number of genes and exons involved, as well as the per gene frequency of new sequence insertions should be specified when stating: "Overall, we found 73,982 alleles within genes (Fig S14), of which 18,095 were within exons."

We thank the reviewer for this suggestion. We have added the additional information to the result section:

“Overall, we found 73,982 alleles within 7644 genes (9.68 alleles/gene) (Fig S22), of which 18,095 were within 3772 exons in 3112 genes (5.81 alleles/gene) (Fig 5B). In particular, 1,383 alleles overlap 306 OMIM (Amberger et al. 2015) exons in 275 OMIM genes (5.03 alleles/gene) that were previously associated with Mendelian diseases and phenotypes (Fig 5C). When binning these alleles by frequency, the majority were singletons and rare variants. Singleton SVs accounted for 51,733 SVs in 6638 genes (7.79 alleles/gene), 13,083 SVs within 2932 exons in 2530 genes (5.17 alleles/gene) and 978 SVs within OMIM exons in 242 OMIM genes (4.04 alleles/gene).”

11. Please define the term "singleton" when referring to SVs.

We have edited the mentions of singletons to mention “that were observed only once” when first used in the context of alleles or SVs.

12. The sentence, "In comparison, singleton SVs were slightly less common in genic regions (69.2%) and much less common in intergenic regions (62.7%)." is difficult to understand and requires further clarification.

Indeed, the numbers in brackets may be ambiguous. We have edited the text to clarify:

“In particular, 72.4% of SVs in exons and 70.7% of SVs in OMIM exons were singletons. In comparison, singleton SVs were slightly **less represented** in genic regions (69.2% **of SVs were singletons**) and much less represented in intergenic regions (62.7% **of SVs were singletons**).”

13. A comparison of the KMT2E deletion to SV calling from short-read GS with multiple callers is necessary. Also, it would be helpful to include the methods used prior to HiFi GS and how the data was analyzed. Please incorporate a screenshot from IGV of srGS data in figure 6.

The family was included in the GA4K study in 5/2020 and had no earlier genetic testing. Initial analyses performed in 2020 included whole exome sequencing, which did not reveal pathogenic variation. This was followed by reanalyses using whole genome short-read sequencing in 2021 where SNVs were prioritized for analyses using machine learning tools and rare SVs were annotated by AnnotSV (pipeline described in Genet Med. 2022 Jun;24(6):1336-1348.) where rare SVs overlapping annotated OMIM genes were followed up. However, the KMT2E DEL (called by Dragen 3.6.3) in the patient was not flagged for follow-up since the OMIM gene field was missing from AnnotSV likely due to late addition of the gene to OMIM (original report of KMT2E related disease or O'Donnell-Luria-Rodan syndrome/OMIM:618512, was in 2019). This family was then routed to HiFi-GS where the diagnostic, previously unprioritized variant was detected among minigraph variants associated with high phrank score and subsequently clinically confirmed and reported to the family. We note that the new short-read annotation pipeline (Illumina Emedgene 32.0.19 released 06/2023) that incorporates SNVs and SVs, and which we now used in reanalyses of srGS VCFs appropriately flags the KMT2E DEL variant among top ten (“most likely”) candidates for this family. Finally, uncovering this previously unreported pathogenic variant by prioritizing a small number of rare SVs in a minigraph pangenome validates the principle of assembly-based discovery of rare structural variation. We added this information to the results section and the IGV screenshot of srGS data to supplementary data (**new Fig S25**):

“While the *KMT2E* variant is not exclusive to minigraph (and is validated by short-read GS, Fig S25) it had not previously been prioritized for follow-up and clinical validation (see Table S1 for testing history).”

Reviewer #2 (Remarks to the Author):

The authors investigate using pan genome graphs for rare disease analysis with a particular interest in SVs. The idea is quite interesting since the allele frequency spectrum that applies to rare disease seems to be far below what has been possible with most pangenomes to date. They did haploid assembly on 287 proband, and with 94 additional haploid assemblies they created a pangenome. From that they found SVs. Every additional genome they added provided 500 additional sequences, which, as they point out, means that many more alleles are to be discovered. It also means that this data set will not be great for determining frequencies for alleles that are on the rarer end of the spectrum, which is directly at odds with the author's objective.

We apologize that the objective of our study wasn't sufficiently clear in the first version of our manuscript. Our objective was to do an in-depth analysis of structural variants, enabled by a pangenome approach, in a large rare disease cohort. We acknowledge that our sample size remains too small to generalize our frequencies to a population but our identification of rare SVs is still directly relevant to the probands being profiled. However, this sample still allows us to remove obviously common variation and prioritize those SVs that are more likely to be truly rare. To clarify this point, we have modified to the title to be more explicit about our focus on SV:

"Pangenome graphs improve the analysis of **structural variants** in rare genetic diseases"

We also modified the abstract to say:

"These data allowed us to build a population graph genome representing a unified SV callset in GA4K, identify common variation and **prioritize SVs that are more likely to cause genetic disease (MAF < 0.01).**"

We also did an additional analysis using the HGSVC assemblies that are not part of the graph to estimate how many of our singletons are not in fact singletons due to sampling error:

"As a result, we found that the HGSVC assemblies contain 17.0% of the HPRC singletons but only 4.85% of the GA4K singletons, indicating that the sampling error is smaller in the GA4K population."

We also added the following clarification on the objective of the study in the introduction:

"Here, we explore the benefits of using such a strategy to **characterize structural variation in a rare disease cohort, exclude common non-pathogenic or infrequent (MAF 1-5%) variation and prioritize SVs that are sufficiently rare to be causal.** Also, we show how pangenome methods can be used along with other tools to improve sensitivity and specificity in detecting **SVs.**"

Finally, we also acknowledge some of the limitations in the discussion as well as emphasizing the utility of such a pangenomic resource:

“We released a useful resource to exclude common variation and keep SVs that are much more likely to be rare, even if the SV frequencies in this sample may not generalize to all populations. Moreover, our resource allows users to expand this graph genome with their own assemblies, enabling rare SV discovery in any assembly .”

My main issue with this paper is the focus on rare SVs and the small sample size. I think they could (but don't) make the case that even with the small sample size, if performance improves on the relatively rare SVs then we should expect that to continue to be true for truly rare variants. But they never make that case. The one section of the paper that addresses improving rare SV calling is also confusing. There is this hypothesis “We hypothesized that a consensus of reference-based and assembly-based methods would improve the precision of rare SVs over reference-based methods alone.” I do not really see how the following experiment tests this hypothesis.

We agree with the reviewer that the analysis we provided of SV replication did not clearly show the improvement we observed. We have reworked the section to directly compare the precision of reference based SV calling results against the precision of consensus-based SV calling to show the improvement of precision from 38% to 56% (all SVs) and 11% to 61% (rare SVs), and changed the terminology to match reader expectation, referring to the consensus (reference+assembly based) SV calls instead of the concordant set, and providing the precision of all the reference-based SV calls (previously this would have been the union of the concordant and the discordant sets). We edited the text accordingly:

“Then, to test if the proportion of false positives relative to true positives was reduced by such an approach in GA4K, we used an independent set of Illumina srGS Manta (18) SV calls (Methods) as the source of truth, and investigated the precision of replicating common and rare (<5% MAF) SVs against this truth set. Reference-based SV calls alone are replicated with a precision of 38% across all frequencies, while reference-based rare SV calls are replicated with a precision of 11%. However, if we consider the consensus of reference-based PBSV calls and assembly-based minigraph calls, SVs of all frequencies are replicated with a precision of 56%, and the rare SV subset is replicated with a precision of 61% (Table S5), indicating that combining assembly and reference-based methods improves the precision of rare SV calls significantly.”

We further tested the hypothesis by benchmarking on HG001. We compared the PBSV and HG001 consensus against the GIAB calls and found that precision increases to 94.8%, which is 4.9 % more than PBSV alone (89.9%). We added these results to the **new table S4** and the text:

“First, we confirmed our expectations by benchmarking on the HG001 GIAB truth set (Table S4), where the consensus SV set showed higher precision while being 17.1% smaller than minigraph and 11.1% smaller than PBSV. “

We also note in discussion:

“We established higher precision of rare SV consensus calls where reference and minigraph based variants are concordant, which is particularly important for SVs outside known disease genes where the prior probability of pathogenicity is lower. High quality rare SV catalog among undiagnosed cases will form basis for new disease gene discovery.”

I am troubled by the statement “As expected, singleton alleles were the most likely to be called from misassembled sequences, while very common alleles were the least likely” Rare disease research is about singleton alleles. Since the signal we care the most about is also the most troubling, I would like the authors to expand on this point. How many singletons were found? How many were called from misassembled sequences? How do we know they are misassembled? I want to be reassured that we are not throwing away important sources of variation. I think right now they are just excluded.

The reviewer is right that rare disease research is about singleton alleles, which is why we conservatively validated our assemblies with Flagger in order to exclude mis-assemblies disguised as singleton SVs. At the same time, precision is important since false positives due to missassembly would overwhelm the clinical prioritization of SVs downstream. The validation with Flagger automatically labels sequences as invalid if the observed read depth over an assembled sequence deviates strongly from the expected value, which may exclude true SVs due to variation in the read depth. Moreover, the reviewer asks for more details on the exact counts of validated and invalidated singleton SVs, and the manner of validation. In total, we found 215578 singleton SVs, of which 172697 were validated and 42881 (19.9% of all singleton SVs) were marked as unreliable. This said, validation with Flagger is a conservative statistical approach that clusters regions by coverage. It is possible that some sequences are labeled as collapsed or duplicated when in fact they are not. We changed “misassembled” to “unreliable” and added these points to the results:

“As expected, singleton alleles that were observed only once were the most likely to be called from unreliable sequences, with 42,881 of 215,578 singleton alleles (19.9%) being rejected by Flagger, while very common alleles were the least likely to be rejected (Fig S3)”

“we count up to 560 alleles in the same locus” Wow! There are only 668 haplotypes. Please comment on what this locus is and what is known about it.

The reviewer correctly observes that this locus (CHM13v2 chr14:30135899-30138115) shows a very large number of alleles. In CHM13v2, it is annotated as a ~3 kbp simple tandem repeat with a unit length of 27 bp. To learn more, we plotted the allele lengths of this simple tandem repeat as a histogram and set the bin width to 27 bp to match the unit length (**new Fig S4**). We see that their length ranges from a few copies to more than 200 copies and averages 96 copies. In particular, this distribution appears to feature many modes. Such a distribution could be the result of a naturally unstable repeat that has many frequent forms, each one contracting and expanding to generate a gaussian distribution around its length. At the same time, other nested

polymorphisms or small errors in assembly could add additional variation around each mode. At the same time, all these alleles pass validation with Flagger, suggesting that any potential missassemblies in this locus are not sufficiently large to create a deviation from the expected read depth at any point over their length. Moreover, this locus is also highly multi-allelic in the HPRC pangenome, where they observe 65 alleles in 94 genomes. Finally, we note that the number of loci featuring an extremely large number of alleles is a small fraction of all polymorphisms (Fig 1D).

Fig S4: Distribution of allele lengths in a highly polymorphic simple tandem repeat with a unit length of 27bp (CHM13v2 chr14:30135899-30138115) with 560 observed alleles.

We edited the result section to reflect these caveats:

“As expected, the few loci with extremely large numbers of alleles are unstable simple and short tandem repeats, which naturally create many alleles but are difficult to align and require additional analysis (Fig S4).”

I liked the experiment that used the twins to determine false positive/negative rates. The language needs to be cleared up here because you cannot differentiate between a false positive and a false negative, which prevents conclusions around true positive rate etc. Specifically, I need some clearer mathematical justification that statements such as “Thus, we consider increased allele sharing between siblings to be evidence of lower false positive and negative rates.” and “The increase in allele sharing between siblings suggests that the SV calls obtained

with minigraph have a lower false positive and false negative rate.” are true. I would expect that some of these SVs could be genotyped in the parents, so the authors may want to look at mendelian violations to help determine FP and FN.

We agree that the allele sharing analysis cannot distinguish between false positive and false negative events. We do think allele sharing should increase with lower error rate since correctly genotyping a shared allele in one sibling while failing in the other sibling would appear as an allele that is not shared. More precisely, given a genotyping error rate e , and assuming that genotyping errors happen independently in the two siblings, the probability for a **shared allele** to be correctly genotyped in both siblings is $P(e) = (1 - e)(1 - e)$ where e takes values between 0 and 1. $P(e)$ is monotonic over this interval and we see that $P(e)$ is lowest where the error rate is 0 and highest where the error rate is 1.

Also, the suggestion to check FP and FN rates through Mendelian violations was very useful. We pursued this in a trio where parents and proband featured HiFi assemblies and summarized the results in the **new Table S3**. In these results, minigraph shows fewer Mendelian violations and demonstrates better false positive and false negative rates.

We added the following text to the results:

“While allele sharing indicates a lower overall error rate, it cannot distinguish between false positive or false negative errors. Thus, we checked Mendelian violations in the GA4K232 trio (Methods) in an attempt to disambiguate these two types of errors and found that minigraph has a lower false positive rate and false negative rate relative to PBSV (Table S3).“

We also amended the Methods:

“We assessed Mendelian violations on the GA4K232 trio, where parents and proband featured HiFi assemblies and PBSV calls. We used SURVIVOR as before to merge the PBSV trio genotypes. We defined Mendelian violations as genotypes that are impossible given the proband and parental genotypes. When an allele is homozygous in both parents, the proband must also be homozygous. If only one of the parents is homozygous, the proband must be at least heterozygous. Otherwise, we consider these events to be false negatives. Genotypes where the proband has more copies of an allele than the parental genotypes allow is a false positive event.”

In general, it would be nice if the discussion about the SV landscape broke the analysis up in the SV types. I would expect the numbers to be better for DEL and worse for INS, and better for big things and worse for little things. Summing things up into just one number is a little too high-level.

We agree that different SV types pose different challenges during genotyping, with deletions being easier to call than insertions. This should result in different frequency spectra. For instance, the frequency spectrum of SVs that are more difficult to detect should be skewed towards rarer counts, since more events would be missed across our genomes. To check this,

we stratified the frequencies of GA4K-only SVs by the type of bubble in which they are found (complex, insertion, deletion, methods for full description) in the **new Fig S20**. We found that complex SVs, which are found in multi-allelic bubbles, are the most skewed towards rare alleles, followed by insertions, with deletions being the least skewed towards rare alleles. This agrees with our initial expectations, since correctly calling a complex SV requires precisely disambiguating between multiple alleles, calling an insertion requires fully assembling across the region, while a deletion requires the least amount of information. Moreover, we see that longer alleles are skewed towards rarer frequencies (Fig 4C), which could be explained by a higher chance of assembly gaps over these longer intervals.

Fig S20: SV frequency spectrum stratified by type.

We added these new observations to the results section:

“Then, we checked if complex, insertion or deletion GA4K-only SVs follow different frequency distributions (Fig S20). We found that complex SVs are the most skewed towards rare alleles, followed by insertions and then deletions.”

Given that assembly quality is such an issue, I would suggest moving those figures from the supplement to the main paper. This is one of the most important results of this paper.

As described above, we added the new panels Fig 1A-B to describe the quality of assemblies in the first main figure and the results.

Reviewer #3 (Remarks to the Author):

Summary The manuscript “Pangenome graphs improve the analysis of rare genetic diseases” describes a study investigating a large cohort of 287 trios with rare pediatric disease from the Genomic Answers for Kids (GA4K) program. These probands from GA4K had undergone PacBio HIFI sequencing, and the authors took a unique approach for analysis of this dataset by building graph genomes for each case. They compared structural variant (SV) yield from the graph genomes to more traditional reference-based methods. The authors also incorporate 94 control haploid genomes released by the Human Pangenome Reference Consortium (HPRC) to better filter for rare variants. Examination of the functional SVs discovered from the graph found over 1,000 rare exon disrupting events. They further narrow this list to 30 SVs with potential disease relevance including a pathogenic deletion in KMT2E. Overall, I found the approach taken in this paper to be intriguing and certainly think the results are of interest to the field. The introduction does a fantastic job explaining why graph genomes could be of interest to the clinical genetic community and the benchmarking performed is mostly adequate. My greatest area of concern is the rare disease analyses, which overstates the utility of the graph method since the authors fail to perform an adequate comparison of pathogenic yield with both short read and ref-based long read methods. Based on the presented results it doesn’t seem graph genomes actually improve the analysis of rare disease since the pathogenic variants found were also discovered in short reads. I don’t think this is a huge problem as the utility of genome graphs is likely to come in interpretation of difficult to map regions, which currently cannot be easily interpreted. I have the following comments to help strengthen the manuscript.

We agree with the reviewer in that the utility of graph genomes will likely be found in difficult to map regions where interpretation is currently difficult. As mentioned above, we have now made several changes to the manuscript to emphasize that graph genomes improve the analysis of structural variation, as opposed to directly improving the analysis of rare diseases. First, we have change the title to explicitly focus on structural variants:

“Pangenome graphs improve the analysis of **structural variants** in rare genetic diseases”

Second, we now emphasize in the abstract the value of the unified SV callsets across many genomes that are provided by graph genomes and the filtering and prioritizing of SVs that genome graphs enable:

“These data allowed us to build a population graph genome representing a **unified SV callset in GA4K, identify common variation and prioritize SVs that are more likely to be rare.**”

The same points are also further elaborated in the introduction:

“**While tools exist to call and cluster SVs (9–11)**, they rely on heuristics such as the maximum distance between events **or proximity graphs (12)**, which may erroneously split or merge SVs because they do not consider the entire genome assembly or **variation between alleles in complex loci**. Therefore, accurate estimation of SV allele frequency may benefit from tools that **align and compare the various alleles observed in complex loci.**”

“Pangenome graphs also provide a unified SV callset where the boundaries of polymorphisms are delimited by bubbles and the alleles are precisely defined as paths through bubbles. This allows for more robust allele frequencies, especially in the case of multiallelic SVs.”

Major Comments 1. There needs to be a clearer description of what prior genetic results went into filtering this cohort since that will have a dramatic impact on the expected diagnostic rate. It seems clinical sequencing was performed but is that whole genome or whole exome? Were only cases with pathogenic SNVs removed but what about SV. Given SV are the focus of this paper it seems important to distinguish whether those have been investigated. The overall rare disease yield found in the study low so I am guessing additional SV filtering must have been applied.

We have now included the **new Table S1** that summarizes the prior genetic results that were conducted in this cohort. In this table, we show which samples were analyzed with exome, whole genome sequencing or chromosome microarray sequencing. Therefore, a subset of imbalanced structural variation was necessarily considered during chromosomal microarray analysis. However, the resolution of this method is worse than genome sequencing. EWS and WGS data was systematically screened only for SNVs prior to producing the assemblies for this cohort. Systematic analysis of SVs was released on 03/2023 and was not used before assembling the genomes.

We edited the results to clarify this:

“In this cohort, **prior to assembling the genomes**, more than 90% of the probands remained undiagnosed **after chromosomal microarray analysis** or even standard clinical sequencing and systematic exploration of putatively causative single nucleotide variants (**Table S1**), with only less than 10% being eventually diagnosed.”

2. The Iso-Seq analysis could be really interesting but is poorly described in the main text. Please add more detail such as the number of sites investigated and across how many samples. Also, what tissue was investigated and does the RNA seem to be coding or noncoding?

We have now expanded the text in our isoseq analysis description to highlight the three tissue types assayed, the assembly and alignment of the transcripts and the genome-wide coverage assessment of the isoseq alignments to the hifiasm hap1 and hap2 assemblies in 500bp tiled bins genome-wide.

We edited the results to now describe the isoseq methods as follows:

“For one of these trios, in addition to the srWGS data, we also have Iso-Seq RNA expression for three different cell types (blood, iPSC, neuronally-differentiated iPSC),

which we aligned to the paternal and maternal personal genomes of the proband. Barcoded primers were removed with PacBio's demultiplexing tool *lima*, followed by *isoseq refine* to assemble full-length non-chimeric transcripts (FLNC). Assembled FLNC transcripts were aligned to the hifiasm hap1 and hap2 assemblies of the proband using gapped minimap2. Mosdepth (38) was used to determine binned (500 bp tiles) coverage across the contigs for each of the assemblies.”

3. Benchmarking with long read data is challenging given a lack of truth data for hard to map regions. In cleaner non-repetitive genomic regions though short read WGS should be adequate for benchmarking. The authors perform this benchmarking but after already describing much dataset. I recommend incorporating this earlier in the benchmarking section of the results.

Extensive benchmarking on long read data has been done in Liao et al (2023) (Liao et al. 2023), where they show excellent recall and precision in easy genomic regions for SVs, and more moderate performance in challenging repetitive parts of the genome that at least matches reference-based methods. Since our goal is to characterize SV using existing methods in these rare disease genomes, we think the existing level of benchmarking in this dataset is appropriate and that the performance is in line with previous expectations. We now add these remarks in the benchmarking results section:

“Over all genotypes, minigraph achieves a recall 0.78 of and a precision of 0.80 against PBSV **which is in line with previous benchmarking of these methods (Liao et al. 2023) in difficult regions of the genome.**”

4. Assessing shared variants across family members can provide a rough measure of genotype accuracy but is still fraught with issues. For instance, a false positive SV that is an assembly error across many samples would be more likely to look shared between sibs even though false. A permutation approach building a distribution from random sib pairs could help protect against some of these technical biases and provide a better comparison between PBSV and the graph genome approach.

We agree with the reviewers that permuting the sibling pairs is needed to get a better understanding of the expectation, especially since SURVIVOR and minigraph merge SVs in the same locus in very different ways. After permuting the siblings, the minigraph shows an allele sharing of 7.5%. We have added the new permutation analysis to the results section and the **new figure S15** and emphasize that these results are affected by the two different SV merging strategies used by SURVIVOR and minigraph:

“**When randomly permuted sibling pairs, minigraph show an average 7.5% more allele sharing (Fig S16). While allele sharing indicates a lower overall error rate, it is affected by the different SV merging strategies employed by the two methods and cannot distinguish between false positive or false negative errors.**”

SV sharing between permuted sibling pairs

Fig S15: Allele sharing between randomly permuted sibling pairs in GA4K, calculated with PBSV and SURVIVOR versus minigraph.

5. The authors clearly demonstrate the ethnicity of the HPRC are widely different from the GA4K, which would influence rare variant filtering. Given the majority of rare pathogenic variants fall in clean genic regions could the author not use the short read SVs from large-population cohorts (e.g, gnomAD) to further filter the GA4K during disease assessment.

Thank you for the suggestion. To evaluate the effect of filtering out GA4K SVs that are found in gnomAD, we overlapped all GA4K singleton alleles with gnomAD-SV, a SV resource compiled from short reads sequencing data. Of the 132,391 singleton GA4K alleles, 94,875 SVs overlap a SV region in gnomAD-SV. However, even if a GA4K allele overlaps a gnomAD-SV, we cannot know if it's the same allele, since gnomAD SVs are not sequence resolved. We think this is particularly important, since SVs are often multiallelic. Therefore, this result shows that a large number of GA4K singletons occur at a low frequency or merely in the same SV hotspots as in the broader population. We added these results to the results section:

“Of the GA4K singletons, 94,875 SVs overlap a gnomAD-SV interval (Collins et al. 2020). Since gnomAD-SV are not sequence resolved, this indicates that most singletons may either exist at a low frequency or occur in the same SV hotspots as in the broader population.”

6. The title of the paper is “Pangenome graphs improve the analysis of rare genetic diseases”, but the current analysis doesn’t do a proper assessment to allow for this statement. The lack of comparison between graph and ref-based methods (both long and short read) for the diagnostic analysis is glaring. It is really important to show whether the graph-based method adds any additional yield over the more traditional methods.

This point was also brought up by the other reviewers. Please see our answers above, where we edit the title, abstract, introduction and discussion to be more explicit about the actual benefits of genome graphs in the GA4K assemblies.

7. This study serves as a proof of concept of the power of graph genomes but until we have a better functional annotation of newly discovered regions these methods will not be fully appreciated. I feel this is needs to be mentioned in the discussion.

We agree with the reviewer that the functional annotation of newly discovered human regions is an important step in revealing new DNA with clinical significance. We have added these points to the discussion:

“Moreover, our efforts to annotate the structural variation in these assemblies require an overlap with the existing functional annotation of the reference genomes. Better annotation of newly discovered regions would likely help identify more DNA with clinical significance and further increase diagnostic yield.”

8. The abstract and title should be revised to better reflect the actual result of the study which shows minimal additional diagnostic utility under current clinical guidelines.

The reviewer correctly notes that we focused more on creating a resource that makes use of new emerging panenomic methods to create a unified SV callset across many genomes and to enable filtering out common SVs in order to focus on SVs that are more likely to be rare. Further advances in SV annotation would be required to drastically increase the diagnostic rate in the GA4K genomes. Thus, we have changed the title to reflect our emphasis on the analysis on structural variants in rare genetic disease as opposed to being a paper that implies to increase diagnostic yield:

“Pangenome graphs improve the analysis **of structural variants in** rare genetic diseases”

We also changed the abstract to clarify the objective of the study:

“These data allowed us to build a population graph genome representing a unified SV callset in GA4K, identify common variation and **prioritize SVs that are more likely to cause genetic disease (MAF < 0.01).**”

At the same time, we emphasize in the introduction the ability of pangenome graphs to create a unified SV callset across many genomes that enables to accurately disambiguate alleles within complex loci:

“Here, we explore the benefits of using such a strategy to **characterize structural variation in a rare disease cohort, exclude common non-pathogenic or infrequent (MAF 1-5%) variation and prioritize SVs that are sufficiently rare to be causal.**”

Minor Comments 1. I am confused by this sentence “We independently confirmed some of the unanchored contigs for one offspring via coverage from mapped srWGS sequencing reads from that sample and its parents, and validate that contigs inherited from one parent (where there was high coverage from the offspring and only one parent) were unambiguously almost always inherited only from the appropriate parent, (Table S1), indicating that these unplaced portions of the pangenome graph can be additional, potentially de novo, family-inherited DNA sequence.”

I don't think anything can be said about de novo mutation here. Of course, any variant type can be de novo but seems very strange to highlight here given that is not really being looked at since the parents don't have long read sequencing.

We agree with the reviewer that the unanchored pangenome sequence could be but not necessarily de novo. We edited the results accordingly:

“indicating that these unplaced portions of the pangenome can be additional **non-reference**, family-inherited DNA sequence.”

3. Heat map seems bimodal but hard to compare with heat map. Replacing or supplementing with something like a violin plot could better show the distribution of the precision and recall.

The heatmap currently shows cumulative distributions for both recall and precision. We took a second look at this distribution but we do not see any modes that are larger than the top one. The cumulative distributions rise smoothly, until they reach the large mode that lies at >80% precision and recall. We now added the violin plots to the supplements and reference them in the results to facilitate the interpretation of the heatmap (new Fig S12A-B):

“This yields a two dimensional distribution describing the recall (**Fig S12A**) and precision (**Fig S12B**) for each minigraph SV in each sample, which we visualize as a heatmap (Fig 3A).”

Fig S12: A) Recall and B) precision density distribution of minigraph genotypes against PBSV for figure 3A.

4. I would like to see per sample variant counts for PBSV and more detailed description of that callset given its importance to the study.

Indeed, it might be interesting to see how many calls PBSV makes relative to minigraph over the wider set of probands, even if we have this information for the twin pair. We calculated the per sample variant counts for PBSV in the **new figure S17C**:

Fig S17C: Number of SVs called by PBSV per diploid genome.

We added the additional information to the supplements and the results section:

“On average, we genotyped 18,326 non-reference SVs **per haplotype (Fig S17A)**, or **28,261 SVs per diploid genome (Fig S17B)**, a figure that is in line with previous findings (6) but that is also influenced by assembly quality (Fig S18) and genome diversity. **At the same time, PBSV calls 22,428 SVs per diploid genome (Fig S17C).**”

5. Which specific OMIM genes were used in the genes of interest analysis? A focus on autosomal dominant may help better understand false positive rate since we wouldn't expect very many disruptions of those.

We used the full set of OMIM genes in the dataset, and did not restrict it in any way. We found rare SVs with these genes, then the results were ranked with Phrank and then manually curated to create Table S6 which lists variants with pathogenic potential. Based on additional comments from other reviewers we now discuss exciting new and alternative methods to do the graph-based SV calling in the Discussion (see comments re minigraph-cactus and PanGenome Graph Builder), but agree with the reviewer that a comprehensive comparison and evaluation of graph SV calling methods with curated validation sets would be extremely interesting.

6. The ACOX1 inversion doesn't appear to be exon disrupting and therefore likely has no predicted functional consequence.

The reviewer is right in that the breakpoints of the inversion are not within exons. However, it may still impact the expression of the gene. Given that ACOX 1 is expected to cause disease through gain of function, it may still have functional consequences if the inversion increases the expression of this gene. Thus, expression assays need to be conducted in order to support or exclude the functional consequences of this variant. We address this in the results section:

“Also, a disease candidate inversion involving *ACOX1* locus was observed **that rearranges several exons. However, typically dominant ACOX1 mutations are gain-of-function, and therefore follow-up RNA expression studies are required (Fig S24).**”

7. Figure legends S15 and S16 are messed up

We thank the reviewer for catching the swapped captions. We have corrected this.

8. I don't understand the definition of “highly polymorphic (non-constrained) exons”. Is the exon disrupted by many SVs so you don't think it is evolutionary constrained? Is this some sort of measure from gnomAD?

The reviewer correctly interprets our phrasing. While it is not a measure from gnomAD, we use the number of SVs observed in these exons over the entirety of our cohort as a proxy to check if an exons is evolutionarily constrained. We clarify this in the text:

“Among the rare SVs impacting the resulting 40 filtered exons, 10 were seen in highly polymorphic exons, **suggesting that they are not evolutionary constrained**, or mapped to non-OMIM genes.”

9. Fail to see why this method may miss translocations. Please describe in more detail in the discussion.

Minigraph specifically filters out aligned contigs with ends that map to different chromosomes or that contain events that exceed 100 kbp (Li, Feng, and Chu 2020). To capture such events, we would need to use tools such as PGGB that can capture such events but are more expensive to run (Erik Garrison et al. 2023). We now address these points in the discussion:

“Such a progressive method is efficient but it depends on the order of genomes that are incorporated and might miss events such as translocations (Leonard et al. 2023) **because minigraph removes sequences mapping to multiple chromosomes or events that are longer than 100 kbp.**”

Reviewer #4 (Remarks to the Author):

In this study, Groza et al use long-read (HiFi) sequencing on probands in 287 parent-offspring trios (Illumina on the parents). A pangenome graph was created on the assemblies of the probands along with the HPRC genomes. The pangenome graph was used to compare variants across individuals. A significant finding is transcripts mapping back to nonreference unanchored contigs. An expected finding is that combining minigraph+pbsv calls increases precision.

The major comment is that it was never quite clear just what the benefit was of using the pangenome graph. For example, it does not appear that an effort was made to create a unified callset using modern approaches for merging calls (e.g. Jasmine <https://www.nature.com/articles/s41592-022-01753-3>). Furthermore, minigraph is in fact less accurate than simply calling variants using assembled sequences and dipcall, particularly for determining breakpoints.

The benefit of using the pangenome graph was indeed to create a unified call set of SVs that is defined in terms of bubbles and paths through these bubbles, similarly to what the reviewer suggests we should do with Jasmine. While Jasmine seems to employ more sophisticated heuristics than SURVIVOR such as proximity graphs, it does not seem to align every allele at a locus to the previously observed alleles in other genomes, which minigraph does. We expect that merging SVs in an allele-aware manner as in minigraph would increase our ability to

distinguish highly similar SV alleles in complex multi-allelic loci. We now comment on this in the introduction:

“Moreover, it remains difficult to compare alleles between genomes, since the genomes are only related to the reference genome and not to each other. While tools exist to call and cluster SVs (9–11), they rely on heuristics such as the maximum distance between events **or proximity graphs (12)**, which may erroneously split or merge SVs because they do not consider the entire genome assembly **or variation between alleles in complex loci.**”

In terms of clarifying our objectives, we have now modified the title, abstract, introduction and discussions. See detailed responses above.

Lastly, minigraph used to lack base level alignments which made the location of breakpoints less accurate. However, since v.0.17, minigraph features base level alignments (with the new `-c` argument) that are taken into consideration during graph generation (<https://github.com/lh3/minigraph/releases/tag/v0.17>), which might improve the precision of the breakpoints. Indeed, HPRC recently released a new version of their minigraph-cactus graph since the original publication that uses this feature (<https://github.com/ComparativeGenomicsToolkit/cactus/blob/master/doc/mc-pangenomes/hprc-v1.1-mc.md#hprc-version-11-minigraph-cactus-release>). Still, we did not do any benchmarking to compare the accuracy of breakpoints between dipcall, minigraph-cactus or previous versions of minigraph. We address this future direction in the discussion:

“Furthermore, this pangenome may be extended in the future by adding base-level variation with minigraph-cactus (28), which would reveal any small nested variation that may exist within structural variants and refine SV breakpoints.”

Finally, a lot of researchers have been hoping that long read sequencing will help discover new causal variants. This study has the potential to answer how pangenomes increase diagnostic yield, but instead a high level approach was taken to quantify an increase in precision using graph+pbsv. The effect on diagnostic yield should be noted, first by detecting pathogenic variants in the short read data either with Manta, or a method that is much more precise (Delly, lumpy, and wham all have greater precision, which despite the drop in sensitivity, are more useful for detecting pathogenic variants than Manta). Then a comparison with a single method (such as pbsv, or better dipcall on the assembled sequences).

We agree that this study has the potential to show how pangenomes may increase diagnostic yield. However, the study was not set up to directly increase diagnostic yield. Our intent was to use novel pangenomic techniques to characterize structural variants in this particular set of rare disease assemblies. Moreover, some barriers remain downstream of pangenomic analysis, such as the annotation of the functional impact of structural variation, particularly in complex loci. So far, assemblies and pangenomes may precisely disambiguate a multi-allelic SV, but the means to ascertain its clinical impact are still manual and very time consuming.

The reviewer also requests a comparison with a more traditional method. Thus we compared minigraph calls against chromosomal microarray (CMA) results that were available for a subset of the sample. On average, we detect most CMA regions in minigraph (mean 79.5%, median 100%). Therefore, the diagnostic yield of minigraph should be at least similar to methods already used in the clinic.

Fig S13: Fraction of Chromosomal Microarray (CMA) SV calls reproduced by minigraph across CMA samples.

We added these points to the discussion and result section:

“Moreover, our efforts to annotate the structural variation in these assemblies require an overlap with the existing annotation of the reference genomes. Therefore, it is possible that methods that automatically annotate human de novo assemblies at scale could identify more sequences with clinical significance and increase diagnostic yield.”

“A similar sensitivity is also achieved when comparing to chromosomal microarray (CMA) results where minigraph recalls on average 79.5% (median 100%) of CMA SVs in each sample (Fig S13, Methods).”

The rest of the comments are minor.

Methods: the HPRC used a combination of minigraph and cactus to refine breakpoints around structural variants, this may be tried (or noted).

We agree that expanding this pangenome with base-level variation is a future next step for this collection of assemblies using minigraph-cactus. However, from our conversation with the developers of cactus, there may be an upper limit to the number of genomes that is below the number of genomes of our cohort. This is due to a technical limitation in one of the file formats used (HAL). This would need to be rectified before we proceed. We added this note in the discussion:

“Furthermore, this pangenome may be extended in the future by adding base-level variation with minigraph-cactus (25), which would reveal any small nested variation that may exist within structural variants.”

How many of the rare SVs are near tangles (VNTR sequences) in the pangenome graph?

The reviewer asked an informative question about the distribution of rare structural variation relative to other loci that are rich in SVs. To answer this, we defined a tangle to be a polymorphic locus with more than 2 alleles (not bi-allelic) and obtained 27,246 such tangles. Then we calculated the distance of each singleton SV to these tangles. We found that 77.4% of all singleton SVs (HPRC and GA4K) are within 100 bp of one of these tangles. We have added the following to the results:

“Of the 185,926 singleton SV alleles that were observed only once, 77.4% occur within 100 bp of a polymorphic locus with more than 2 alleles.”

Minor: the very long novel alleles (>50kb) are worth more consideration. First, try remapping the novel allele + some flanking sequence to CHM13 T2T (or just hg38) to make sure these are not artifacts of minigraph alignment. Next, it would be good to know which genes map to these novel sequences.

We have taken a closer look at the very large alleles that exceed 50kbp. In total, we counted 14,632 long alleles that are specific to GA4K distributed among 766 bubbles. We realigned the alleles to chm13v2, with the flanking source and sink nodes included, then examined a sample manually in IGV. The alignments do show evidence of large structural variation in at least one of the alleles of the locus, which is sufficient to create a large bubble even if most long alleles in the locus have minor variations on each other. We noticed three patterns. First, many large alleles are contractions or expansions of tandem repeats (i.e D4Z4 shown below in the **new figure S18**). Second, inversions of large regions of the genome create large alleles (i.e ACOX1 example in the manuscript). Lastly, sometimes a small DNA segment replaces a much larger DNA segment and creates a short allele and a much longer allele, similarly to a deletion. Thus, these large alleles tend to be genomic rearrangements in various configurations. We have added these details to the results:

“The very long alleles are created by expansions and contractions of tandem repeats (Fig S19), inversions, or when large sequences are replaced by a much smaller sequence in one of the genomes.”

Fig S18: Long D4Z4 repeat alleles in GA4K can range from 10 kbp up to 100 kbp in length.

Minor: enrichment (or depletion) analysis should be done for singleton SVs in genes/exons/OMIM exons.

This is a relevant analysis looking to see if the frequency of singleton SVs is different between various parts of the genome. In fact, we have made an attempt at such an analysis in Fig 5D, where the allele frequency spectra show that singleton SVs are more frequent in genic/exonic/OMIM regions relative to intergenic regions. For example, we state in the results that 72.4% of SVs in exons are singleton versus only 62.7% in intergenic regions (Fig 5D). This is consistent with selection removing structural variation in functional regions and skewing the frequency distribution towards lower counts. We edited the results to introduce this analysis more explicitly:

“Next, we checked if singleton alleles are enriched or depleted between intergenic, genic or exonic regions of the genome.”

Minor: what is the percent reduction in calls when combining minigraph + pbsv?

In our new HG001 benchmarking results described in the **new table S4**, PBSV called 22613 SVs and minigraph called 24239 SVs. After the consensus, 20104 variants remained, giving a 17.1% reduction over minigraph alone and 11.1% reduction over PBSV alone. The resulting consensus callset had a higher precision than PBSV or minigraph when compared to the GIAB HG001 SV truth set.

“First, we confirmed our expectations by benchmarking on the HG001 GIAB truth set (Table S4), where the consensus SV set showed higher precision while being 17.1% smaller than minigraph and 11.1% smaller than PBSV.”

Minor: generally, the calls based on de novo assemblies (using dipcall) are superior to pbsv calls. For the assemblies that are high quality (>30x input coverage), it would be good to redo some of the joint calling analysis using dipcall variants instead of pbsv. Furthermore, it's unclear whether the benefit was inherent to use of the graph itself, or simply a separate variant calling approach.

We agree with the reviewer that minigraph and dipcall both benefit from calling structural variation from an assembly as opposed to a collection of aligned long reads to varying degrees. At the same time, Liao et al. 2023 showed that graph approaches outperform workflows that only rely on GRCh38. However, while an increase in performance is great, analyzing SVs in rare disease genomes encounters other challenges such as creating unified sets of SVs and distinguishing between many SV alleles in complex loci. During this study, we appreciated how minigraph provides these seamlessly through its definition of polymorphism as bubbles and alleles as paths. With dipcall, we would need to perform additional steps, with tools that may collapse similar but different alleles in complex loci using heuristics based on proximity of SVs. We make note of this in the introduction:

“While SV callers that operate on whole genome assemblies exist (7, 8), their approach of comparing a proband genome against a single reference genome may fail, even with high quality genome assemblies, since some regions may be absent or present very different alleles. Moreover, it remains difficult to compare alleles between genomes, since the genomes are only related to the reference genome and not to each other.”

“While tools exist to call and cluster SVs (9–11), they rely on heuristics such as the maximum distance between events or proximity graphs (12), which may erroneously split or merge SVs because they do not consider the entire genome assembly or variation between alleles in complex loci.”

Minor: The low mapping rate of non-reference nodes is puzzling, even if they are not annotated as repeat nodes. In the past, I've tried to see what insertion sequences (e.g. equivalent to non-reference nodes) have no match in the genome using: 1. repeat masking of sequences, then mapping back to the genome, then mapping back to nr. Almost everything had some decent alignment.

The reviewer observes that we imply that non-reference sequences with repeats and the non-reference sequences mapping to the genome are mutually exclusive. However, this was not our intention, since sequences consisting of repeats map to their copies in the reference genome. Thus, the mapping rate is in fact 98.7 percent, which we calculate by excluding the 1.3% of sequence that is not repeat masked or mapped to the genome. We have edited the relevant section to make this clear:

“Next, we aligned the non-reference nodes to the CHM13v2 reference and found that 24.5% of all non-reference sequences (8.9% in HPRC and 15.6% in GA4K-only) were not repeats but mapped to some region in the genome (Fig 2C), suggesting duplications or other rearrangement events. **Overall, 98.7% of all non-reference sequence mapped to repeats or other parts of the genome, leaving 1.3% (7.5 Mbp) of the pangenome as putatively novel sequences.**”

Minor: it would be good to start with a figure describing the dataset: distribution of coverage, assembly quality, etc. This will ground the readers in the quality of the dataset.

The reviewer along with other reviewers is interested in a more detailed summary of the dataset quality. For this purpose, we have added the subpanels (new Fig 1A-B) describing the distribution of sequencing depth and assembly N50:

Fig 1: A) Distribution of HiFi sequencing depth across the proband genomes. B) Distribution of assembly contiguity (N50) across the proband diploid assemblies.

These are accompanied by the following new text in the result section:

“at a mean depth of 27X (Fig 1A, median 27X, range 6X-48X)”

“obtaining a mean N50 of 18.2 Mbp (Fig 1B, median 16.4 Mbp, range 78.6 kbp-55.3 Mbp)”

In addition, we also included a more detailed summary of the validation results that were obtained with Flagger in the new Fig S1 shown below and added a citation in the relevant

results section:

“To ensure all genotypes were derived from reliably assembled sequences, we validated the assemblies with Flagger (Liao et al. 2023) and excluded the genotypes supported by collapsed, duplicated or low coverage regions (**Fig S1**, Methods).”

Fig S1: Flagger results summarizing the proportion of each assembly that was labeled by Flagger as collapsed, duplicated, erroneously assembled, or properly assembled (haploid).

Use uniform processing of the HPRC assemblies to calculate the number of SV in OMIM genes. The HGSVC assemblies may be added for improved sampling of low allele frequency.

We agree with the reviewer that some singletons may not be in fact singletons due sampling error. As suggested, we ascertained the extent of sampling error by genotyping the HGSVC assemblies against our genome graph. Overall, 17.0% of singleton alleles HPRC were found in HGSVC. In contrast, only 4.85% of GA4K singleton alleles were found in HGSVC. These results indicate that the sampling error in our 287 proband genomes is lower than in HPRC, which is a higher quality but smaller collection. These results also show that even larger collections of assemblies are needed to confidently identify very rare alleles. We have added these results to the main text:

“To ascertain how many singletons are due to sampling error, we genotyped the 88 haplotype resolved HGSVC assemblies (6) against our genome graph. As a result, we found that the HGSVC assemblies contain 17.0% of the HPRC singletons but only 4.85%

of the GA4K singletons, indicating that the sampling error is smaller in the GA4K population.“

REVIEWERS' COMMENTS

Reviewer #1 (Remarks to the Author):

The manuscript is much improved and all my comments have been addressed

Reviewer #2 (Remarks to the Author):

The authors nicely addressed all of my issues.

In addition, I think that the authors have addressed reviewer 3's concerns.

Reviewer #4 (Remarks to the Author):

My comments have been addressed. Reading the other reviewers comments, the authors have undertaken thorough analysis of their data.